# Interleaved Selective State Space Models for Efficient WiFi-Based 3D Multi-Person Pose Estimation

**Quang-Anh N.D.** [1]  **Kok-Seng Wong** [1] [2]

## Abstract

WiFi-based human pose estimation offers privacy-preserving and occlusion-robust sensing, but current Transformer-based approaches suffer from quadratic complexity and lack explicit inductive biases for the structure of Channel State Information (CSI). We propose WiFi-Mamba, the first State Space Model (SSM) architecture for WiFi-based 3D multi-person pose estimation. Our approach introduces three key contributions: (1) a Dual-Stream Selective SSM that processes amplitude and phase through parallel pathways with cross-stream state coupling to respect their distinct physical properties, (2) Selective State Attention for pose query decoding with SSM-derived sequential context, and (3) Persistent SSM Memory for temporal consistency across frames without recurrent memory explosion. Extensive experiments on the Person-in-WiFi 3D dataset, covering both single-person and multi-person, demonstrate a 16-27% MPJPE reduction across varying numbers of persons while using only 4.4% of the baseline parameters (2.14M vs. 48.2M), achieving superior efficiency-accuracy trade-offs particularly beneficial for edge deployment in privacy-sensitive continuous monitoring scenarios.

## 1. Introduction

With the global population aged 60+ projected to reach 2.1 billion by 2050 (World Health Organization, 2022), the demand for continuous health monitoring is surging. Human Pose Estimation (HPE) is essential for fall detection and rehabilitation tracking (Xu et al., 2025; Sedaghati et al., 2025), yet traditional camera-based systems face significant privacy concerns in sensitive areas (e.g., bedrooms, bathrooms,

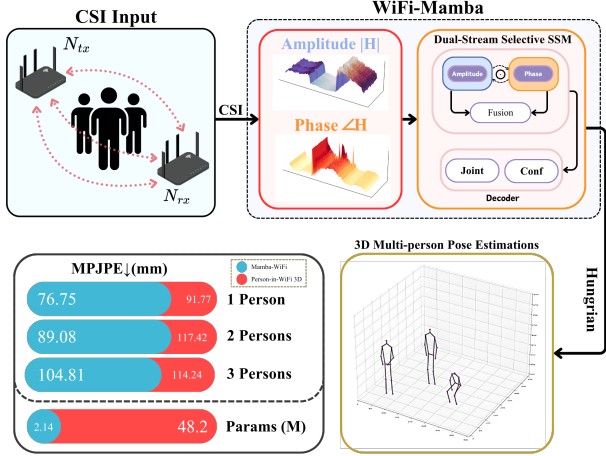

*Figure 1.* `WiFi-Mamba` **for 3D multi-person pose estimation.** WiFi routers capture CSI encoding human motion. Our dual-stream architecture processes amplitude and phase through selective SSMs with cross-stream coupling, achieving linear complexity $\mathcal{O}(n)$. WiFi-Mamba achieves 16-27% MPJPE improvements across scenarios while using only 4.4% of baseline parameters.

nursing facilities, etc.) and are hindered by lighting and occlusions (Wang et al., 2019; Jiang et al., 2020). WiFi-based sensing addresses these limitations by leveraging the ubiquitous wireless infrastructure. By analyzing Channel State Information (CSI), which characterizes the fine-grained frequency response of wireless channels, these systems perceive human motion through walls and in darkness while preserving visual privacy (Miao et al., 2025; Wei et al., 2025). The non-intrusive nature of WiFi sensing makes it particularly suitable for monitoring elderly individuals who may forget to wear devices or feel uncomfortable with camera surveillance (Muaaz et al., 2022). However, identifying HPE from CSI presents substantial challenges. First, CSI manifests as high-dimensional, complex-valued tensors in the frequency domain, with complex inter-subcarrier correlations (Halperin et al., 2011; Wang et al., 2017; Xie et al., 2019). Additionally, the mapping from CSI to pose is highly nonlinear because amplitude reflects signal attenuation due to body shadowing. At the same time, phase encodes velocity-induced Doppler frequency shifts from limb velocities (Yang et al., 2022).

[1]Center for Environmental Intelligence, VinUniversity, Hanoi, Vietnam [2]College of Engineering and Computer Science, VinUniversity, Hanoi, Vietnam. Correspondence to: Kok-Seng Wong <wong.ks@vinuni.edu.vn>.

*Proceedings of the $43^{rd}$ International Conference on Machine Learning*, Seoul, South Korea. PMLR 306, 2026. Copyright 2026 by the author(s).

Moving from single to multi-person 3D pose estimation compounds these challenges at the signal level. Since CSI is the coherent superposition of all propagation paths (Halperin et al., 2011; Wang et al., 2017), each additional person contributes reflections that linearly mix at every receive antenna (Yue et al., 2018; Zeng et al., 2020), producing an *entangled* signal rather than a separable per-person signature, with body-to-body shadowing further distorting individual attenuation patterns. This entanglement manifests empirically as severe feature overlap between persons (Custance et al., 2026), leading the authors of Person-in-WiFi 3D (Yan et al., 2024) to report that a naive 3D extension of their single-person pipeline failed to converge, motivating a Transformer-based end-to-end decoder. While this resolves the assignment problem, the quadratic complexity $\mathcal{O}(n^2)$ of self-attention (Katharopoulos et al., 2020) and the lack of inductive biases for WiFi signal physics, yield parameter-heavy models unsuitable for resource-constrained edge deployment where low latency and privacy are essential (Wang et al., 2017; Yang et al., 2023a).

State Space Models (SSMs) have recently been presented as a promising alternative for efficient sequence modeling. Vision Mamba (Zhu et al., 2024) demonstrates that SSM-based architectures can achieve $2.8\times$ faster inference and 86.8% GPU memory reduction compared to Transformers in image understanding. In contrast, Video Mamba (Li et al., 2025) shows strong capability in capturing long-term temporal dynamics. However, these vision-oriented methods assume regular grid structures. They cannot handle the unique characteristics of complex-valued frequency-domain WiFi-CSI signals with inter-subcarrier dependencies, which require a fusion of amplitude and phase modalities that encode distinct physical properties. To address these challenges, we propose **WiFi-Mamba**, a novel SSM architecture design for WiFi-based 3D multi-person pose estimation.

Our contributions can be summarized as follows:

● We propose **WiFi-Mamba**, a novel SSM-based architecture for WiFi-based pose estimation, built on a Dual-Stream Selective State Space Model (DS$^3$M) that processes amplitude and phase in parallel pathways, with cross-stream state coupling to account for their distinct physical properties.

● We introduce Selective State Attention (SSA) for enhanced pose query decoding, leveraging SSM-derived sequential context and Persistent SSM Memory to maintain temporal consistency across frames without recurrent memory explosion.

● We conducted extensive experiments on the Person-in-WiFi 3D dataset and achieved state-of-the-art results across varying numbers of persons, using only 4.4% of the baseline parameters, establishing a new efficiency-accuracy trade-off for practical deployment.

## 2. Related Work

### 2.1. 3D Multi-Person Pose Estimation

Multi-person pose estimation follows two paradigms, including (i) top-down methods that first detect individuals and estimate poses later (Fang et al., 2017; Wang et al., 2021; Xiao et al., 2018), and (ii) bottom-up methods that detect all joints simultaneously and group them into individuals (Cao et al., 2017; Cheng et al., 2020). Beyond these traditional approaches, transformer-based methods have shown a third paradigm, with DETR (Carion et al., 2020) enabling end-to-end detection without hand-crafted components like non-maximum suppression, inspiring pose-specific adaptations. Building on this, PETR (Shi et al., 2022) formulates multi-person pose estimation as hierarchical set prediction using learnable pose queries. Following this direction, QueryPose (Xiao et al., 2022) and ED-Pose (Yang et al., 2023b) used auxiliary detection tasks for improved query initialization, while GroupPose (Liu et al., 2023) unifies pose and keypoint decoders for cross-instance interaction. For 3D HPE, MVGFormer (Liao et al., 2024) explicitly models multi-view geometry within Transformer decoders. However, these transformer-based methods suffer from quadratic computational complexity $\mathcal{O}(T^2)$ with respect to sequence length $T$, limiting their scalability for continuous monitoring applications.

### 2.2. State Space Models for Signal Processing

SSMs originate in control theory (Kalman, 1960), and S4 (Gu et al., 2022) demonstrates effective modeling of long-range dependencies. Mamba (Gu & Dao, 2024) introduced selective state spaces with input-dependent transitions, achieving linear complexity $\mathcal{O}(L)$, further improved by Mamba-2 (Dao & Gu, 2024) by improving the efficiency through structured state space duality, enabling $2\text{-}8\times$ faster training and inference compared to the original Mamba while maintaining comparable accuracy. These properties make SSMs suited for pose estimation, as selective mechanisms allow adaptive filtering of motion-relevant signals from environmental noise, while implicit temporal modeling via hidden states naturally enforces temporal consistency without relying on explicit recurrent gating.

Recent applications to wireless signal processing include MAMCA (Zhang et al., 2024) for modulation classification, CPMamba (Luo et al., 2025) for MIMO-OFDM channel estimation, IQUMamba (Gao et al., 2026) for blind source separation, and Radar-Mamba (Gao et al., 2025) for millimeter-wave point cloud enhancement. Also, Tan et al. (2025) uses bidirectional Mamba to WiFi-based activity recognition, achieving high accuracy with reduced parameters. However, existing SSM applications for wireless sensing focus on single-task classification or signal enhancement and do not address multi-person 3D pose estimation from WiFi-CSI.

## 2.3. WiFi-Based Pose Estimation

WiFi sensing utilizes CSI for device-free perception (Miao et al., 2025; Wei et al., 2025), evolving from activity recognition (Wang et al., 2015a) and gesture detection (Pu et al., 2013) to fine-grained pose estimation. Person-in-WiFi (Wang et al., 2019) pioneered 2D pose estimation, WiPose (Jiang et al., 2020) extended to 3D, and Dense-Pose from WiFi (Geng et al., 2022) achieved dense correspondence comparable to image-based methods. Recent Transformer-based approaches include MetaFi (Yang et al., 2022), MetaFi++ (Zhou et al., 2023a), and CSI-Former (Zhou et al., 2023b). The state-of-the-art Person-in-WiFi 3D (Yan et al., 2024) adopts a DETR-style architecture for end-to-end multi-person 3D pose estimation, achieving high accuracy but requiring 48.2M parameters. Moreover, these Transformer-based methods typically incur quadratic computational complexity, which limits their scalability for long-duration or continuous scenarios, and they lack explicit mechanisms to maintain temporal consistency across frames, relying instead on implicit data-driven learning. To the best of our knowledge, no prior work has explored SSMs for efficient sequential modeling in WiFi-based HPE.

## 3. Methodology

### 3.1. Problem Setup

We address the problem of multi-person 3D human pose estimation from WiFi-CSI. Given a CSI sequence $\mathcal{X} \in \mathbb{R}^{N_{tx} \times N_{rx} \times N_{ant} \times N_{sub} \times T}$, captured by a WiFi system with $N_{tx}$ transmitters, $N_{rx}$ receivers, $N_{ant}$ antennas, $N_{sub}$ Orthogonal Frequency Division Multiplexing (OFDM) subcarriers, and $T$ time steps, our goal is to estimate 3D joint positions $\mathcal{P} = \{P_i\}_{i=1}^{N_p}$ for $N_p$ persons, where each $P_i \in \mathbb{R}^{J \times 3}$ denotes the 3D coordinates of $J$ anatomical keypoints.

WiFi-based 3D pose estimation poses challenges distinct from those of camera-based methods (Wang et al., 2021; Fang et al., 2017). CSI encodes human motion implicitly through multipath propagation (Wang et al., 2019; Jiang et al., 2020). The received CSI at subcarrier frequency $f$ and time $t$ can be modeled as:

$$H(f,t) = \sum_{k=1}^{K} \alpha_k(t) e^{-j2\pi f \tau_k(t)} + n(f,t), \quad (1)$$

where $\alpha_k(t)$ and $\tau_k(t)$ denote the attenuation and delay of the $k$-th path, and $n(f,t)$ is additive noise. In addition, the pose-to-CSI mapping is highly non-linear and sensitive to environmental factors (Yang et al., 2022). Unlike images with strong spatial locality, CSI signals lie in the frequency domain and exhibit complex inter-subcarrier correlations induced by the OFDM physical layer (Halperin et al., 2011; Xie et al., 2019; Wang et al., 2017), making effective pose modeling particularly challenging. This formu-

lation in Eq. (1) shows two key observations that motivate our design: (1) CSI contains both amplitude $|\alpha_k|$ and phase $2\pi f \tau_k$, which encodes motion information, and (2) the temporal evolution across frames $t$ carries dynamics essential for pose reconstruction. These insights directly inspire our preprocessing strategy and dual-stream architecture.

### 3.2. Preliminaries

Before detailing each component, we first walkthrough the mathematical foundations of Selective State Space Models (S6) that underpin our architecture. The S6 (Gu & Dao, 2024) extends classical Linear Time-Invariant (LTI) systems (Gu et al., 2021) via input-dependent parameterization. A continuous-time SSM maps an input signal $u(t) \in \mathbb{R}$ to an output $y(t) \in \mathbb{R}$ through a latent state $h(t) \in \mathbb{R}^N$:

$$h'(t) = \mathbf{A}h(t) + \mathbf{B}u(t), \quad (2)$$
$$y(t) = \mathbf{C}h(t) + \mathbf{D}u(t), \quad (3)$$

where $\mathbf{A} \in \mathbb{R}^{N \times N}$, $\mathbf{B} \in \mathbb{R}^{N \times 1}$, $\mathbf{C} \in \mathbb{R}^{1 \times N}$, and $\mathbf{D} \in \mathbb{R}$ are system parameters, and $N$ denotes the state dimension. For discrete-time processing, the continuous system is discretized using the Zero-Order Hold (ZOH) method (Gu et al., 2022) with a learnable step size $\Delta$:

$$\bar{\mathbf{A}} = \exp(\Delta \mathbf{A}), \quad (4)$$
$$\bar{\mathbf{B}} = (\Delta \mathbf{A})^{-1} (\exp(\Delta \mathbf{A}) - \mathbf{I}) \Delta \mathbf{B} \approx \Delta \mathbf{B}, \quad (5)$$

where the approximation holds for small $\Delta$ and is commonly adopted for computational efficiency. This yields the discrete-time recurrence:

$$h_t = \bar{\mathbf{A}} h_{t-1} + \bar{\mathbf{B}} u_t, \quad (6)$$
$$y_t = \mathbf{C} h_t + \mathbf{D} u_t. \quad (7)$$

The main contribution of S6 (Gu & Dao, 2024) is to make the parameters $\mathbf{B}$, $\mathbf{C}$, and the discretization step size $\Delta$ functions of the input, enabling content-aware and adaptive sequence modeling. Specifically, given an input representation $x_t$ at time $t$ obtained after linear projection and convolution, these parameters are defined as:

$$\mathbf{B}_t = \text{Linear}_B(x_t), \quad \mathbf{C}_t = \text{Linear}_C(x_t), $$
$$\Delta_t = \text{softplus}(\text{Linear}_\Delta(x_t)). \quad (8)$$

This selective mechanism is particularly relevant for WiFi sensing, for the model can learn to amplify state updates when detecting motion-induced CSI variations while suppressing updates during static periods. Also, SSMs suit WiFi sensing through input-dependent parameters $\mathbf{B}_t$, $\mathbf{C}_t$, $\Delta_t$ that distinguish motion from static reflections, linear complexity $\mathcal{O}(L)$ for efficient long-sequence processing, and hidden states that maintain temporal context without memory explosion, and for multi-person estimation, the

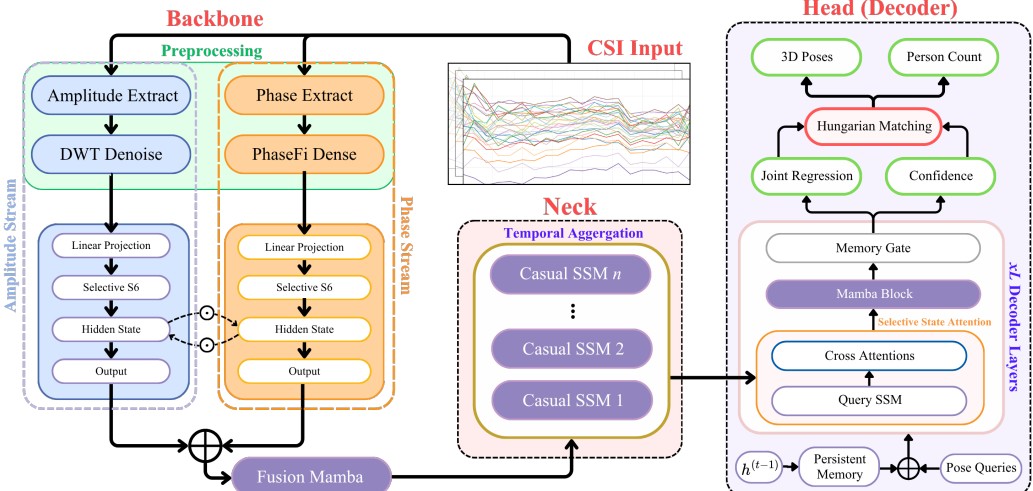

*Figure 2.* Overall architecture of `WiFi-Mamba`. **Backbone:** Preprocessing extracts amplitude and phase from CSI input, followed by denoising. Each stream processes through Linear Projection and Selective S6 blocks with hidden states coupled via cross-stream gating, where $\odot$ denotes element-wise multiplication. Outputs are concatenated ($\oplus$) and fused via Fusion Mamba. **Neck:** Three causal SSM layers aggregate temporal information. **Head (Decoder):** Selective State Attention applies SSM-based query contextualization before cross-attention. Persistent Memory ($h^{(t-1)}$) maintains temporal consistency via the Memory Gate and the Mamba Block. Prediction heads output 3D poses, confidence scores, and person count, with Hungarian matching for multi-person assignment.

backbone processes shared CSI, formal complexity analysis is provided in Appendix E. At the same time, the decoder employs multiple pose queries inspired by DETR (Carion et al., 2020). We reuse S6 across specialized components where DS³M processes amplitude/phase streams, Fusion Mamba unifies features, the neck aggregates temporal information, and the decoder contextualizes queries.

### 3.3. Overview of WiFi-Mamba

Building upon the observations from Eq. (1), CSI contains amplitude and phase components encoding distinct motion information. That temporal evolution carries essential dynamics, which we propose `WiFi-Mamba` is specifically designed for WiFi-CSI-based 3D multi-person pose estimation. As illustrated in Fig. 2, the complete WiFi-Mamba architecture comprises three main components:

- **Backbone:** Processes raw CSI through a preprocessing pipeline (Section 3.4) that extracts and denoises amplitude and phase, followed by DS³M layers (Section 3.5) for dual-stream encoding, and fusion Mamba blocks for unified representation learning.

- **Neck:** Aggregates temporal information using causal SSM layers, transforming encoded CSI features into a sequence-aware representation.

- **Head:** Employs learnable pose queries processed through decoder layers utilized with Persistent SSM Memory (Section 3.7) and Selective State Attention (Section 3.6), culminating in prediction heads.

This architecture achieves linear complexity $\mathcal{O}(n)$ while maintaining the ability to capture long-range dependencies through selective state space mechanisms, addressing the limitations of quadratic-complexity Transformers for continuous WiFi sensing applications.

### 3.4. Physics-Informed CSI Preprocessing

Raw CSI measurements are corrupted by Carrier Frequency Offset (CFO), Sampling Frequency Offset (SFO), and environmental interference. We apply a physics-informed preprocessing pipeline (See Fig. 2) to extract and denoise the amplitude and phase components identified in Eq. (1).

**Amplitude and phase extraction.** CSI data is inherently complex-valued, typically stored as concatenated real and imaginary components. Given raw CSI data , we extract the amplitude and phase by Eq. (9):

$$A[k] = \sqrt{\text{Re}[k]^2 + \text{Im}[k]^2},$$
$$\phi[k] = \text{atan2}(\text{Im}[k], \text{Re}[k]) \tag{9}$$

where $A[k]$ and $\phi[k]$ denote the amplitude and phase of the $k$-th subcarrier, with values in $\mathbb{R}^+$ and $[-\pi, \pi]$ respectively, preserving the physical properties of the wireless signal.

**Denoising.** We denoise amplitude via a learnable Discrete Wavelet Transform (DWT) with soft thresholding:

$$\tilde{d}_l = \text{sign}(d_l) \cdot \max(|d_l| - \tau_l, 0) \tag{10}$$

where $\tau_l = \text{softplus}(\theta_l)$ is a learnable threshold for level $l$. For phase, we apply PhaseFi (Wang et al., 2015b) to remove

CFO/SFO-induced linear distortion:

$$\phi_{\text{corrected}}[k] = \phi[k] - (\alpha k + \beta) \qquad (11)$$

where $\alpha$ and $\beta$ are estimated via linear regression, followed by learnable residual refinement, detailed implementation and visualizations are presented in Appendix C. By obtaining clean amplitude and phase streams that preserve the complementary physical properties identified in Eq. (1), we now introduce a dual-stream architecture that processes these modalities through parallel SSM pathways while allowing information exchange via cross-stream coupling.

### 3.5. Dual-Stream State Space Model

A common intuition in WiFi sensing literature is that CSI amplitude captures body shadowing while phase encodes Doppler shifts from limb velocity. From the CSI propagation model in Eq. (1), both amplitude and phase are functions of the same underlying path parameters $\alpha_k(t)$ and $\tau_k(t)$, and are therefore not strictly physically separable. A more rigorous motivation for processing them independently lies in their *statistical asymmetry*. Amplitude is a strictly non-negative quantity with a smooth, skewed distribution shaped by multipath superposition (Tse & Viswanath, 2005), whereas phase is a signed periodic quantity over $[-\pi, \pi]$ and is highly susceptible to hardware-induced CFO and SFO, which introduce per-subcarrier offsets and linear rotation errors (Zhuo et al., 2017). These differences make shared parameterization ill-suited, as denoising and state-update dynamics tuned for one modality generally transfer poorly to the other, this motivates the DS$^3$M, which processes amplitude and phase through parallel SSM streams (see Backbone in Fig. 2) with a novel cross-stream state coupling mechanism that lets the two streams exchange information while retaining independent parameters.

**Cross-stream state coupling.** Each stream employs the S6 mechanism (Gu & Dao, 2024), computing the discrete-time SSM recurrence from Eq. (6)-(7). The amplitude and phase streams process denoised amplitude and corrected phase features in parallel. Although the two modalities are processed separately to respect their distinct statistical properties, their hidden states encode complementary information that can mutually enhance representation quality. Unlike conventional dual-stream architectures that fuse only final outputs, we couple hidden states directly. Let $h_t^A$ and $h_t^\phi$ denote the hidden states of amplitude and phase streams at time $t$. We compute coupled states as:

$$\tilde{h}_t^A = h_t^A + \lambda \cdot g_t \odot W_{\phi \to A}(h_t^\phi) \qquad (12)$$

$$\tilde{h}_t^\phi = h_t^\phi + \lambda \cdot (1 - g_t) \odot W_{A \to \phi}(h_t^A) \qquad (13)$$

where $\lambda$ is the coupling ratio controlling information exchange, $\odot$ denotes element-wise multiplication, and $W_{\phi \to A}$,

$W_{A \to \phi}$ are learnable projection matrices. The selective gate $g_t$ modulates the direction of information flow:

$$g_t = \sigma\big(W_g[\text{mean}(h_t^A); \text{mean}(h_t^\phi)]\big) \qquad (14)$$

where $\sigma$ denotes the sigmoid function, this gating mechanism learns to balance information exchange adaptively. Specifically, when amplitude indicates high confidence, it contributes more to stabilize phase estimates; conversely, when phase shows consistent motion patterns, it helps disambiguate amplitude fluctuations. Theoretical justification for cross-stream coupling is provided in Appendix G.

**State-based output modulation.** To ensure that state-level interactions propagate to the final representations, the coupled states modulate stream outputs:

$$\tilde{y}^A = y^A \odot \big(1 + \lambda \cdot \sigma(W_m^A \cdot \text{mean}(\tilde{h}_t^A))\big) \qquad (15)$$

and similarly for the phase stream. The modulated outputs are then fused via learned projection by $z = W_f[\tilde{y}^A; \tilde{y}^\phi]$.

The fused representation $z$ captures unified motion information from both modalities and serves as input to subsequent Mamba blocks for further refinement. This representation then flows to the decoder, where pose queries must attend to CSI features to predict joint positions, a process we enhance with SSM-derived sequential context as described next.

### 3.6. Selective State Attention

Standard cross-attention treats query tokens independently, computing attention weights solely from query-key similarity. However, for multi-person pose estimation, queries representing body parts should be contextually aware when attending to CSI features, as the wrist position is constrained by the elbow and related to the shoulder. We propose Selective State Attention (SSA), which enhances cross-attention with SSM-derived sequential context.

**Query contextualization via SSM.** Before computing attention, we process the query sequence through a lightweight SSM to inject sequential context:

$$\tilde{Q} = \text{SSM}(Q), \quad \tilde{Q} \in \mathbb{R}^{K \times D} \qquad (16)$$

where $Q \in \mathbb{R}^{K \times D}$ are the original query representing $K$ candidate poses. Because the SSM propagates information through its hidden state, each contextualized query $\tilde{q}_k$ encodes information from preceding queries $\tilde{q}_{<k}$, introducing explicit inter-query dependency. The contextualized queries $\tilde{Q}$ thus encode relationships among all pose candidates.

**Cross-attention with contextualized queries.** The contextualized queries then attend to the encoded CSI features

via standard scaled dot-product attention:

$$\text{Attention}(\tilde{Q}, K, V) = \text{softmax}\left(\frac{\tilde{Q}K^\top}{\sqrt{d}}\right)V \qquad (17)$$

Unlike standard cross-attention, where per-query logits are computed independently, the SSM-induced coupling in $\tilde{Q}$ means logits vary coherently across query positions, for joint reasoning over shared CSI observations. This is crucial for jointly interpreting queries from different body parts and individuals in multi-person pose estimation. While SSA captures intra-frame spatial relationships, temporal consistency is handled through persistent memory. We address this through persistent memory.

### 3.7. Persistent SSM Memory

Temporal consistency is crucial for pose estimation in sequential models, as human motion is highly continuous, and using this prior can resolve frame-level ambiguities. Traditional approaches employ recurrent networks (*e.g.*, GRU, LSTM) with explicit gating, but these suffer from vanishing gradients over long sequences. We introduce Persistent SSM Memory, which harnesses the selective scan mechanism of SSMs to maintain state information across frames without the memory explosion typical of recurrent architectures.

**Memory architecture.** Our memory mechanism employs $M$ learnable memory tokens $\mathbf{m} \in \mathbb{R}^{M \times D}$ that serve as a compressed representation of historical context. Unlike recurrent networks that use gating, our approach leverages the selective scan mechanism of SSMs. These tokens are prepended to the query sequence:

$$\varphi = [\mathbf{m}; Q] \in \mathbb{R}^{(M+K) \times D} \qquad (18)$$

creating an extended sequence where memory tokens can interact with pose queries through SSM state propagation.

**State persistence across frames.** The combined sequence process is shown by Fig. 2 in Head, it is processed through an SSM started with a hidden state from the previous frame:

$$\tilde{\varphi}, h^{(t)} = \text{SSM}(\varphi, h^{(t-1)}) \qquad (19)$$

where $h^{(t-1)}$ is the hidden state from frame $t-1$. The state $h^{(t)}$ is stored for the next frame, maintaining temporal context across the video sequence. During training, we detach gradients at frame boundaries to limit memory usage while preserving forward temporal information, allowing temporal prior learning without full backpropagation through time.

**Memory-enhanced query output.** The query outputs are extracted from the processed sequence and combined with the original queries via a learned gate:

$$Q_{\text{out}} = g_m \odot \tilde{\varphi}_{M:} + (1 - g_m) \odot Q \qquad (20)$$

where $\tilde{\varphi}_{M:}$ denotes the last $K$ tokens corresponding to queries, and $g_m = \sigma(W_m[Q; \text{mean}(\tilde{\varphi}_{:M})])$ is a memory gate. This gating mechanism learns to balance memory features against fresh query embeddings: when historical context is informative (*e.g.*, continuous motion), the gate increases memory contribution, when the current frame presents novel information, it favors fresh embeddings.

### 3.8. Loss Functions

We use Hungarian matching (Carion et al., 2020; Chen et al., 2022) for bipartite assignment between predictions $\{(\hat{P}_k, \hat{c}_k)\}_{k=1}^{K}$ and ground truth $\{P_i\}_{i=1}^{N_p}$:

$$\pi^* = \text{argmin}_\pi \sum_{i=1}^{N_p} \left( \|P_i - \hat{P}_{\pi(i)}\|_1 - \hat{c}_{\pi(i)} \right). \qquad (21)$$

Given $\pi^*$, the total loss is:

$$\mathcal{L} = \lambda_j \mathcal{L}_{\text{joint}} + \lambda_b \mathcal{L}_{\text{bone}} + \lambda_c \mathcal{L}_{\text{conf}} + \lambda_n \mathcal{L}_{\text{count}}, \qquad (22)$$

where $\mathcal{L}_{\text{joint}}$ supervises matched joint positions with $\ell_1$ loss:

$$\mathcal{L}_{\text{joint}} = \frac{1}{N_p J} \sum_{i=1}^{N_p} \sum_{j=1}^{J} \|P_{i,j} - \hat{P}_{\pi^*(i),j}\|_1, \qquad (23)$$

here, $\mathcal{L}_{\text{bone}}$ enforces anatomical bone length consistency (Pavllo et al., 2019):

$$\mathcal{L}_{\text{bone}} = \frac{1}{N_p |\mathcal{B}|} \sum_{i=1}^{N_p} \sum_{(j_1, j_2) \in \mathcal{B}} \Big| \|P_{i,j_1} - P_{i,j_2}\|_2$$
$$- \|\hat{P}_{\pi^*(i),j_1} - \hat{P}_{\pi^*(i),j_2}\|_2 \Big| \qquad (24)$$

the $\mathcal{L}_{\text{conf}}$ applies Focal Loss on confidence scores with target $y_k{=}1$ for matched queries, and $\mathcal{L}_{\text{count}}$ uses cross-entropy on person count prediction.

## 4. Our Experiments

**Dataset.** We evaluate using the Person-in-WiFi 3D (Yan et al., 2024), the only dataset in our knowledge for WiFi-CSI based 3D multi-persons pose estimation. This dataset contains 72K frames from 9 subjects across 3 indoor environments, with CSI captured from a $3 \times 3$ MIMO setup at 100 Hz, and each spatial stream has 30 OFDM subcarriers. For single-person evaluation, we additionally use the Wi-Pose dataset (Zhou et al., 2022), a benchmark with 166,600 synchronized image-CSI frames from 12 volunteers performing 12 daily actions, collected on 5 GHz 802.11n WiFi at 100 Hz with a $3 \times 3$ antenna setup over 30 subcarriers. The implementation details is presented in Appendix 4.

**Implementation Setting.** The encoder uses 2 DS$^3$M blocks and 2 Mamba blocks, the decoder has 3 SSA layers. Model dimension $d{=}256$, SSM state $N{=}16$, heads

*Table 1.* Single-person pose estimation on Person-in-WiFi 3D dataset. WiFi-Mamba vs. state-of-the-art methods. Lower is better for error metrics. Best results are in **bold**, second-best are underlined.

| Method | Params (M) | MPJPE | PA-MPJPE | MPJDLE (h) | MPJDLE (v) | MPJDLE (d) | PCK@30 | PCK@20 | PCK@10 |
|---|---|---|---|---|---|---|---|---|---|
| MetaFi++ (Zhou et al., 2023a) | 26.5 | 116.40 | 72.40 | 56.94 | 61.01 | 46.36 | 75.31 | 60.45 | 32.91 |
| DT-Pose (Chen et al., 2025) | 4.8 | 92.70 | 60.09 | 41.66 | 50.55 | 38.60 | 81.95 | 71.78 | 50.35 |
| HPE-Li (D. Gian et al., 2025) | **0.8** | 117.70 | 69.90 | 53.64 | 67.19 | 50.06 | 75.27 | 60.35 | 33.78 |
| Person-in-WiFi 3D (Yan et al., 2024) | 48.2 | 91.77 | 64.51 | 37.84 | 51.97 | 45.00 | **94.96** | **86.20** | 59.70 |
| **WiFi-Mamba (Ours)** | 2.14 | **76.75** | **62.85** | **32.11** | **47.77** | **31.41** | 86.29 | 78.07 | **61.09** |

*Table 2.* Single-person pose estimation on WiPose dataset. WiFi-Mamba vs. vs. state-of-the-art methods. Lower is better for error metrics. Best results are in **bold**, second-best are underlined.

| Method | Params (M) | MPJPE | PA-MPJPE | MPJPE (h) | MPJPE (v) | PCK@30 | PCK@20 | PCK@10 |
|---|---|---|---|---|---|---|---|---|
| MetaFi++ (Zhou et al., 2023a) | 26.5 | 58.17 | 35.15 | 32.45 | 39.21 | 53.38 | 36.65 | 13.62 |
| DT-Pose (Chen et al., 2025) | 4.8 | **34.14** | **23.19** | **18.88** | **22.34** | **77.97** | **69.61** | 51.40 |
| HPE-Li (D. Gian et al., 2025) | **0.8** | 39.83 | 25.52 | 22.38 | 26.58 | 70.58 | 56.87 | 33.21 |
| **WiFi-Mamba (Ours)** | 2.14 | 35.82 | 24.06 | 19.66 | 23.62 | 75.92 | 67.75 | **51.99** |

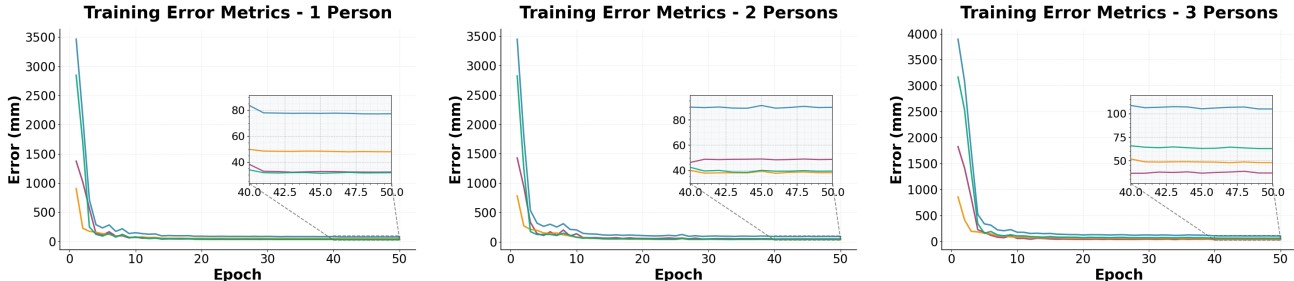

*Figure 3.* Training convergence curves for WiFi-Mamba across one, two, and three-person scenarios.

$H=8$, memory tokens $M=16$. Training uses AdamW with lr$=10^{-4}$ and weight decay $10^{-2}$ for 50 epochs with batch size 16 on a single NVIDIA 4070Ti Super GPU. Coupling ratio $\lambda=1.0$, coupling projections are zero-initialized.

### 4.1. Evaluation Results

**Single-person comparisons.** In this section, we use the same evaluation metrics as Yan et al. (2024), namely Mean Per Joint Position Error (MPJPE) and Mean Per Joint Dimension Location Error (MPJDLE), measured in millimeters across horizontal (h), vertical (v), and depth (d) dimensions, also with PCKs@$\{10, 20, 30\}$ for broader comparison against state-of-the-art methods on WiFi-CSI-based single-person pose estimation. As shown in Table 1, on Person-in-WiFi 3D our method achieves the best performance across all metrics, with 76.75 mm MPJPE, a 16.4% improvement over the Person-in-WiFi 3D baseline (Yan et al., 2024) (91.77 mm) while using only 2.14M parameters vs. 48.2M. The gain is most pronounced in depth (31.41 mm vs. 38.60 mm from DT-Pose (Chen et al., 2025)), validating our cross-stream state coupling for amplitude-phase disambiguation. On the Wi-Pose dataset (Zhou et al., 2022) as in Table 2, WiFi-Mamba reaches 35.82 mm MPJPE, second to

DT-Pose (34.14 mm) with less than half its parameters while outperforming MetaFi++ (Zhou et al., 2023a) and HPE-Li (D. Gian et al., 2025) by 38.4% and 10.1%. Showing that our SSM-based sequential modeling is particularly effective under strict localization tolerances.

**Multi-person comparisons.** In our multi-person comparisons, we use the same evaluation setting as single-person pose estimation. As shown in Table 3, WiFi-Mamba consistently outperforms the baseline across all metrics while using only 4.4% of parameters (2.14M vs. 48.2M). For MPJPE, we achieve 16.4%, 24.1%, and 27.3% reductions in one, two, and three-person scenarios (76.75 vs. 91.77 mm, 89.08 vs. 117.42 mm, 104.81 vs. 144.24 mm), with widening gaps suggesting better handling of multi-person ambiguity. Horizontal and vertical components improve 15.1-25.9% and 8.1-21.7%, demonstrating balanced spatial gains. See Appendix D for qualitative visualization of predictions. Decomposed MPJDLE metrics show depth improvements of 30.2%, 30.9%, and 31.4% for one, two, and three persons, respectively, showing that cross-stream coupling effectively uses amplitude-phase information for depth disambiguation. The training convergence curves are shown in Fig. 3.

## 4.2. Ablation Study

Using the 3-person scenario, we ablate component contributions by comparing amplitude/phase-only vs. coupled processing, persistent memory for temporal consistency, SSA vs. cross-attention, coupling ratio sensitivity, and key architectural hyperparameters.

*Table 3.* Performance comparison between WiFi-Mamba model and Person-in-WiFi 3D model under different numbers of persons.

| Metric | Person-in-WiFi 3D (Yan et al., 2024) (48.2M) | | | WiFi-Mamba (Ours) (2.14M) | | |
|---|---|---|---|---|---|---|
| | 1P | 2P | 3P | 1P | 2P | 3P |
| MPJPE | 91.77 | 121.04 | 144.24 | **76.75** | **89.08** | **104.81** |
| PA-MPJPE | 64.51 | 68.91 | 61.90 | **62.85** | **67.55** | **66.31** |
| MPJDLE (h) | 37.84 | 65.75 | 61.41 | **32.11** | **48.76** | **36.93** |
| MPJDLE (v) | 51.97 | 48.43 | 58.05 | **47.77** | **37.92** | **47.58** |
| MPJDLE (d) | 45.00 | 55.58 | 91.21 | **31.41** | **38.43** | **62.54** |
| PCK@30 | 94.96 | 77.18 | 65.66 | **86.29** | **79.44** | **73.44** |
| PCK@20 | 86.20 | 60.30 | 40.00 | **78.07** | **68.64** | **58.25** |
| PCK@10 | 59.70 | 25.04 | 14.51 | **61.09** | **47.67** | **32.82** |

**Effect of each component.** Table 4 shows ablation results on the three-person scenario. Amplitude-only achieves 119.51 mm MPJPE and phase-only 122.39 mm, confirming both carry meaningful information, with amplitude excelling at horizontal localization (38.48 mm). Dual-stream without coupling achieves 113.71 mm, showing naive fusion captures useful information. However, DS$^3$M with cross-stream coupling achieves 104.81 mm, a 7.8% improvement, validating that respecting distinct statistical properties while enabling controlled state exchange yields superior representations.

*Table 4.* Ablation study on the effect of each component. Lower values are better. **Bold** and underlined values denote the best and second-best results, respectively.

| Method | MPJPE | MPJDLE(h) | MPJDLE(v) | MPJDLE(d) |
|---|---|---|---|---|
| Amplitude Only | 119.51 | 38.48 | 57.80 | 73.69 |
| Phase Only | 122.39 | 41.86 | 57.86 | 72.92 |
| No Separation | 113.71 | 39.58 | 52.96 | 68.14 |
| No Memory | 117.37 | 41.38 | 55.58 | 68.66 |
| Standard Attention | 118.55 | 40.07 | 54.97 | 71.76 |
| **WiFi-Mamba (Ours)** | **104.81** | **36.93** | **47.58** | **62.54** |

Removing persistent memory raises MPJPE by 12.0% (104.81 to 117.37 mm), showing that temporal consistency helps resolve ambiguities. Replacing SSA with standard cross-attention increases MPJPE by 13.1% (104.81 to 118.55 mm), with depth worsening most (62.54 to 71.76 mm). SSA helps because its SSM passes couples queries through a shared hidden state, so each query encodes what earlier queries have attended to prevent the redundant and fragmented attention patterns that independent queries produce when multiple persons compete for the same CSI observations.

**Coupling ratio sensitivity.** Table 5 shows coupling ratio $\lambda$ effects on three-person performance. Without coupling ($\lambda = 0.0$), MPJPE reaches 107.49 mm, 2.6% worse than optimal. Performance improves as $\lambda$ rises from 0.0 to 1.0, achieving the best MPJPE of 104.81 mm at $\lambda = 1.0$. However, higher coupling ($\lambda \geq 2.0$) hurts performance, with $\lambda = 3.0$ reaching 106.99 mm, showing that too-strong cross-stream interaction adds noise. Notably, MPJDLE(d) gains most from proper coupling, improving from 64.72 mm to 62.54 mm, confirming that complementary amplitude-phase information is key for depth disambiguation.

*Table 5.* Coupling ratio $\lambda$ sensitivity on 3-person pose estimation performance. Lower values are better. **Bold** and underlined values denote the best and second-best results, respectively.

| $\lambda$ | MPJPE | MPJDLE(h) | MPJDLE(v) | MPJDLE(d) |
|---|---|---|---|---|
| 0.0 | 107.49 | 36.27 | 49.74 | 64.72 |
| 0.05 | 106.79 | 37.74 | 48.13 | 64.21 |
| 0.1 | 104.98 | 36.75 | 47.72 | 62.79 |
| 0.5 | 105.55 | 38.24 | 49.22 | 61.80 |
| 1.0 | **104.81** | **36.93** | **47.58** | **62.54** |
| 2.0 | 105.37 | 37.70 | 49.69 | 60.17 |
| 3.0 | 106.99 | 49.57 | 49.57 | 63.26 |

**Different architecture configurations.** Table 6 analyzes architecture hyperparameters on three-person scenarios. Deeper encoders reduce MPJPE from 127.33 to 118.04 mm but add 38% parameters. Decoder depth peaks at 3 SSA layers (123.67 mm); fewer lack capacity, more show diminishing returns. SSM state dimension $N \in \{8, 16, 32\}$ performs robustly (119.84-124.21 mm), with memory tokens optimal at $M = 8$ (118.80 mm). All variants underperform the full model (104.81 mm), showing gains stem from synergistic integration, not component scaling.

*Table 6.* Ablation study on architectural hyperparameter ablation. Lower values are better. **Bold** and underlined values denote the best and second-best results, respectively.

| Setting | Params (M) | MPJPE | MPJPE(h) | MPJPE(v) | MPJPE(d) |
|---|---|---|---|---|---|
| **Model depth (DS$^3$M blocks + Mamba blocks)** | | | | | |
| Blocks = 2 | 2.65 | 127.33 | 41.36 | 62.38 | 75.81 |
| Blocks = 4 | 3.66 | 118.04 | 40.56 | 56.99 | 68.61 |
| **SSA layers** | | | | | |
| SSA = 1 | 1.64 | 127.90 | 41.02 | 63.61 | 75.97 |
| SSA = 3 | 2.64 | 123.67 | 41.23 | 60.66 | 72.88 |
| SSA = 4 | 3.14 | 125.49 | 44.00 | 62.16 | 72.49 |
| **SSM state size ablation** | | | | | |
| $N = 8$ | 2.69 | 122.93 | 40.91 | 60.45 | 71.89 |
| $N = 16$ | 2.78 | 124.21 | 39.68 | 60.39 | 76.28 |
| $N = 32$ | 2.96 | 119.84 | 43.27 | 58.19 | 71.36 |
| **Memory tokens ablation** | | | | | |
| $M = 0$ | 2.14 | 118.60 | 42.87 | 57.62 | 68.49 |
| $M = 8$ | 2.14 | 118.80 | 40.96 | 58.75 | 68.61 |
| $M = 32$ | 2.14 | 122.80 | 43.27 | 58.19 | 71.36 |
| **WiFi-Mamba (Ours)** | **2.14** | **104.81** | **36.93** | **47.58** | **62.54** |

*Table 7.* Sensitivity analysis under simulated multipath interference.

| Condition | Paths | Faded (%) | MPJPE | PA-MPJPE | MPJPE(h) | MPJPE(v) | MPJPE(d) | PCK@30 |
|---|---|---|---|---|---|---|---|---|
| Clean | 0 | 0 | 101.36 | 60.72 | 35.91 | 48.24 | 58.14 | 75.9 |
| Mild | 1 | 3 | 111.50 | 64.35 | 37.64 | 52.50 | 65.77 | 72.8 |
| Moderate | 2 | 7 | 167.32 | 73.92 | 75.40 | 68.57 | 82.66 | 60.5 |
| Severe | 3 | 12 | 228.09 | 84.14 | 118.24 | 89.09 | 103.11 | 48.5 |
| Extreme | 5 | 20 | 294.05 | 99.71 | 166.36 | 111.20 | 125.98 | 36.3 |

*Table 8.* Sensitivity analysis under simulated NLOS conditions.

| Condition | Reduction (%) | Phase SNR | MPJPE | PA-MPJPE | MPJPE(h) | MPJPE(v) | MPJPE(d) | PCK@30 |
|---|---|---|---|---|---|---|---|---|
| LOS (Clean) | 0 | $\infty$ | 101.36 | 60.72 | 35.91 | 48.24 | 58.14 | 75.9 |
| Partial NLOS | 10 | 30 | 100.67 | 60.44 | 35.54 | 48.13 | 57.76 | 75.3 |
| Moderate NLOS | 25 | 20 | 105.72 | 61.52 | 36.04 | 51.87 | 60.97 | 72.9 |
| Full NLOS | 45 | 12 | 251.89 | 81.11 | 137.91 | 93.39 | 105.72 | 46.2 |
| Extreme NLOS | 70 | 6 | 625.48 | 130.91 | 373.02 | 184.16 | 282.35 | 19.0 |

**Sensitivity Analysis Under Perturbation** To evaluate robustness, we perturb the CSI tensor under two conditions: multipath interference using frequency-selective fading and delayed reflected paths and NLOS degradation we modeled through amplitude attenuation, per-antenna variation, and SNR-controlled noise.

As shown in Table 7, WiFi-Mamba is reasonably tolerant of mild multipath (1 path, 3% faded with MPJPE 111.50 mm, +10%), but degrades more sharply under moderate-to-severe conditions, with MPJPE reaching 228.09 mm at 3 paths and 294.05 mm at 5 paths. The horizontal component MPJPE(h) is most affected, rising from 35.91 mm to 166.36 mm under extreme interference, consistent with multipath primarily distorting spatial localization. This degradation pattern is physically intuitive, with additional reflected paths introduce constructive and destructive interference across subcarriers, corrupting the amplitude envelope that the dual-stream backbone relies on for spatial disambiguation. Notably, the vertical MPJPE(v) degrades more gradually than MPJPE(h), suggesting that elevation cues encoded in the phase stream are partially preserved even under moderate multipath, as vertical motion induces Doppler shifts that remain distinguishable across a wider range of interference conditions.

Under NLOS conditions as outlined in Table 8, the model exhibits a notable cliff effect. Performance is nearly unchanged through moderate NLOS (25% amplitude reduction, SNR 20dB with MPJPE 105.72 mm), showing the physics-informed preprocessing and dual-stream coupling provide effective resilience at low-to-moderate signal degradation. However, full NLOS (45% reduction, SNR 12dB) causes a sharp collapse to 251.89 mm MPJPE, and extreme NLOS (70%, SNR 6dB) renders pose estimation largely unreliable at 625.48 mm. These results suggest that while WiFi-Mamba handles partial occlusion well, heavily attenuated or reflected signals remain a fundamental challenge, pointing to stronger denoising or antenna diversity as promising directions for future work.

## 5. Limitations and Future Work

The linear recurrent nature of SSMs, while computationally efficient, may limit their ability to capture highly non-local dependencies compared to full attention. Additionally, our evaluation is limited by dataset availability to our knowledge, only one public dataset exists for 3D multi-person WiFi pose estimation, with scenarios with up to three persons. Evaluations on more diverse environments and larger crowds remain future work. Furthermore, FPS cannot be meaningfully computed as throughput depends on hardware-specific CSI sampling and preprocessing pipelines that are not standardized. Future work will establish a real-world evaluation setup with CSI-acquisition settings to enable a systematic comparison of end-to-end latency and efficiency. Beyond benchmarking, we aim to deploy a continuous fall detection system for elder for privacy-preserving sensing.

## 6. Conclusion

We present WiFi-Mamba, the first SSM-based architecture for WiFi-based 3D multi-person pose estimation. Our DS$^3$M with cross-stream state coupling, SSA, and Persistent SSM Memory, achieves 16-27% MPJPE improvements over baselines while reducing the parameter count by 96% from 48.2 M to 2.14 M. Showing that SSMs can effectively capture the complex temporal dynamics of CSI signals with lower computational overhead, establishing a new efficiency-accuracy paradigm.

## Acknowledgements

This work was supported by the Green Serverless Computing for Resource-Efficient AI Training Project at VinUniversity under Grant VUNI.CEI.FS 0002.

## Conflict of Interest Disclosure

Authors have no conflict of interest.

## Impact Statement

This paper advances machine learning methods for temporal modeling of wireless sensing signals. Our work may enable applications in healthcare and smart environments while offering a privacy-friendlier alternative to camera-based systems.

As with other sensing technologies, improper deployment could raise concerns about surveillance and consent, but these issues are well understood and not unique to this work.

We do not foresee any specific negative societal impacts beyond those common to machine-learning-based sensing systems.

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

# A. Abbreviation

*Table 9.* Notation and symbols used throughout the paper.

| Symbol | Description | Symbol | Description |
|---|---|---|---|
| *Input and Output* | | | |
| $\mathcal{X}$ | CSI input tensor | $P_i$ | 3D pose for person $i$, $P_i \in \mathbb{R}^{J \times 3}$ |
| $N_{tx}, N_{rx}$ | Number of Tx and Rx | $N_p$ | Number of persons |
| $N_{ant}$ | Number of antennas | $J$ | Number of joints (keypoints) |
| $N_{sub}$ | Number of OFDM subcarriers | $\mathcal{P}$ | Set of predicted 3D poses |
| $T$ | Sequence length (time steps) | | |
| *CSI Components* | | | |
| $A$ | Amplitude component | $\tau_k(t)$ | Time delay of $k$-th path |
| $\phi$ | Phase component | $K$ | Number of propagation paths |
| $H(f, t)$ | Channel response at $f, t$ | $n(f, t)$ | Additive noise |
| $\alpha_k(t)$ | Attenuation of $k$-th path | | |
| *State Space Model* | | | |
| $h_t$ | Hidden state at time $t$ | $\bar{\mathbf{A}}, \bar{\mathbf{B}}$ | Discretized SSM matrices |
| $u_t$ | Input at time $t$ | $\Delta$ | Discretization step size |
| $y_t$ | Output at time $t$ | $\Delta_t$ | Input-dependent step size |
| $\mathbf{A}, \mathbf{B}, \mathbf{C}, \mathbf{D}$ | SSM system matrices | $N$ | SSM state dimension |
| $x_t$ | Input representation | $D$ | Feature dimension |
| *Dual-Stream Architecture* | | | |
| $h_t^A$ | Amplitude stream state | $W_{\phi \to A}$ | Phase-to-amplitude projection |
| $h_t^\phi$ | Phase stream state | $W_{A \to \phi}$ | Amplitude-to-phase projection |
| $\tilde{h}_t^A$ | Coupled amplitude state | $y^A, y^\phi$ | Stream outputs |
| $\tilde{h}_t^\phi$ | Coupled phase state | $z$ | Fused representation |
| $\lambda$ | Coupling ratio | $W_f$ | Fusion projection matrix |
| $g_t$ | Coupling gate at time $t$ | $W_g$ | Gate projection matrix |
| *Selective State Attention* | | | |
| $Q$ | Query embeddings, $\mathbb{R}^{K \times D}$ | $K$ | Number of pose queries |
| $\tilde{Q}$ | Contextualized queries | $H$ | Number of attention heads |
| *Persistent SSM Memory* | | | |
| $\mathbf{m}$ | Memory tokens, $\mathbb{R}^{M \times D}$ | $h^{(t)}$ | Persistent state at frame $t$ |
| $M$ | Number of memory tokens | $g_m$ | Memory gate |
| $Q_{\text{out}}$ | Memory-enhanced queries | $W_m$ | Memory projection matrix |
| $\varphi$ | Extended memory-query sequence | | |
| *Loss Functions* | | | |
| $\mathcal{L}$ | Total loss | $\mathcal{L}_{\text{conf}}$ | Confidence loss |
| $\mathcal{L}_{\text{joint}}$ | Joint position loss | $\mathcal{L}_{\text{count}}$ | Person count loss |
| $\mathcal{L}_{\text{bone}}$ | Bone length loss | $\lambda_j, \lambda_b, \lambda_c, \lambda_n$ | Loss weights |
| $\mathcal{B}$ | Set of bone pairs | $\pi^*$ | Optimal Hungarian matching |
| *Preprocessing and Other* | | | |
| $L$ | Wavelet decomposition levels | $\rho$ | Spectral radius of $\bar{\mathbf{A}}$ |
| $\tau_l$ | Threshold at level $l$ | $\odot$ | Element-wise multiplication |
| $c_l, d_l$ | Wavelet coefficients | $\sigma(\cdot)$ | Sigmoid function |
| $\theta_l$ | Learnable threshold param | $\|\cdot\|, \|\cdot\|_F$ | Euclidean, Frobenius norm |

# B. Introduction

This supplementary material provides a detailed theoretical analysis supporting the main contributions of WiFi-Mamba. We present details preprocessing process, qualitative results, formal complexity analysis, convergence guarantees, and information-theoretic justifications for our architectural design choices. These theoretical results complement the empirical findings in the main paper and provide principled explanations for the observed performance improvements.

## C. Preprocessing Implementation Details

This section provides complete implementation details for the physics-informed CSI preprocessing pipeline introduced in Section 3.4 of the main paper.

### C.1. Amplitude and Phase Extraction

CSI data is inherently complex-valued, typically stored as concatenated real and imaginary components. Given raw CSI data $\mathbf{H} \in \mathbb{R}^{B \times N \times T \times 60}$, we extract the amplitude and phase by Eq. (9) in the main paper:

$$A[k] = \sqrt{\text{Re}[k]^2 + \text{Im}[k]^2}, \quad \phi[k] = \text{atan2}(\text{Im}[k], \text{Re}[k]) \tag{25}$$

where $A[k]$ and $\phi[k]$ denote the amplitude and phase of the $k$-th subcarrier, respectively. This extraction ensures that amplitude values are strictly non-negative and phase values lie within $[-\pi, \pi]$, preserving the physical properties of the wireless signal.

### C.2. Wavelet-Based Amplitude Denoising

Human motion manifests as relatively low-frequency variations in CSI amplitude, while high-frequency components typically correspond to noise. We employ a learnable Discrete Wavelet Transform (DWT) with soft thresholding to adaptively separate these components.

Using Haar wavelets, we decompose the amplitude signal across $L$ levels by:

$$c_l, d_l = \text{DWT}(c_{l-1}) \quad \text{with } l = 1, \dots, L \tag{26}$$

where $c_l$ and $d_l$ are the approximation and detail coefficients at level $l$, and $c_0 = A$ is the input amplitude signal.

We then apply learnable soft thresholding to the detail coefficients as shown in Eq. (10) of the main paper:

$$\tilde{d}_l = \text{sign}(d_l) \cdot \max(|d_l| - \tau_l, 0) \tag{27}$$

where $\tau_l = \text{softplus}(\theta_l)$ is a learnable threshold parameter for each level. The denoised amplitude is reconstructed via inverse DWT:

$$\tilde{A} = \text{IDWT}(c_L, \{\tilde{d}_l\}_{l=1}^L) \tag{28}$$

yielding a cleaner signal while preserving motion-relevant variations.

Figure 4 illustrates the effectiveness of amplitude denoising at time $t = 10$. The learnable DWT with soft thresholding effectively smooths high-frequency noise (visible as rapid fluctuations in the raw signal) while preserving the underlying amplitude pattern across subcarriers that encodes spatial information about human pose.

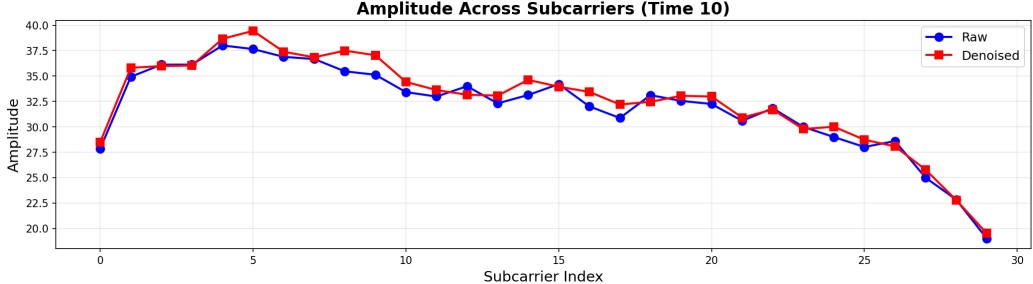

*Figure 4.* Amplitude denoising via learnable DWT. Raw (blue) and denoised (red) amplitude at time $t = 10$. Learnable soft thresholding removes high-frequency noise while preserving motion-relevant signal structure.

### C.3. PhaseFi-Based Phase Denoising

Unlike amplitude, phase measurements suffer from systematic linear distortion caused by CFO and SFO. Following the PhaseFi method (Wang et al., 2015b), we remove this linear trend via least-squares fitting as shown in Eq. (11) of the main paper:

$$\phi_{\text{corrected}}[k] = \phi[k] - (\alpha k + \beta) \tag{29}$$

where $\alpha$ and $\beta$ are estimated via linear regression across subcarrier indices:

$$\alpha, \beta = \arg\min_{\alpha,\beta} \sum_{k=1}^{60} (\phi[k] - \alpha k - \beta)^2 \tag{30}$$

We further refine the corrected phase through a learnable residual transformation to capture any remaining systematic patterns.

Figure 5 demonstrates phase correction effectiveness across multiple time steps. Raw phase measurements (left panel) exhibit clear linear distortion visible as parallel diagonal trends across subcarrier indices a characteristic signature of CFO and SFO. After applying PhaseFi correction (right panel), this systematic bias is removed, revealing the true phase variations induced by human motion.

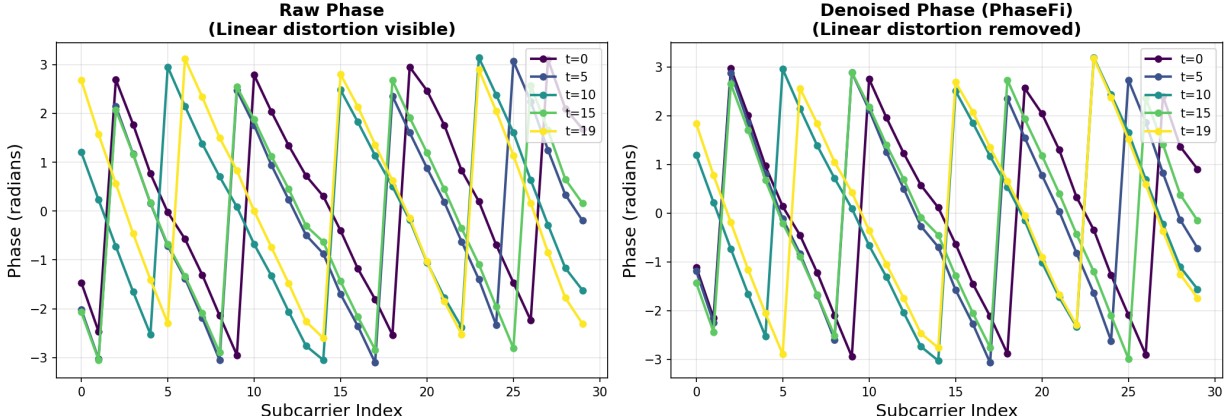

*Figure 5.* Phase correction via PhaseFi across time. **Left:** Raw phase showing linear distortion across subcarriers. **Right:** Denoised phase after removing CFO/SFO-induced linear component via least-squares regression. Five time steps shown ($t = 0, 5, 10, 15, 19$).

Figure 6 shows the complete preprocessing effect at time $t = 10$. The top panel demonstrates amplitude denoising, while the bottom panel illustrates phase correction. The green dashed line represents the estimated linear component ($\alpha k + \beta$ with slope $\alpha = -0.003$) that is removed from raw phase measurements to obtain motion-relevant phase variations.

Confirming that the preprocessing pipeline effectively removes systematic noise and hardware-induced distortions while preserving the complementary motion information encoded in amplitude and phase that is essential for the signals.

## D. Qualitative Visualization

Fig. 7 shows qualitative pose predictions on Person-in-WiFi 3D test set. WiFi-Mamba (red dotted) consistently produces poses closer to ground truth (gray solid) than baseline Person-in-WiFi 3D (teal dashed), on 15 different samples. Notable success on challenging configurations demonstrates effective disambiguation through the proposed architecture.

Also, Fig. 8 further provide qualitative heatmaps of model attention over spatial-temporal CSI features, which show that our design produces more structured, joint-consistent activation patterns than baseline variants, indicating improved feature representation and robustness.

## E. Computational Complexity Analysis

We begin by establishing the computational advantage of WiFi-Mamba over Transformer-based approaches (Vaswani et al., 2017).

**Theorem E.1** (Computational Complexity of WiFi-Mamba). *Following the complexity analysis architecture of Gu & Dao (2024) for selective state space models, we establish that for an input CSI sequence of length $T$ with feature dimension $D$ and SSM state dimension $N$, WiFi-Mamba achieves time complexity $\mathcal{O}(TDN)$ and space complexity $\mathcal{O}(DN)$, compared to $\mathcal{O}(T^2D)$ and $\mathcal{O}(T^2)$ for Transformer-based methods (Vaswani et al., 2017).*

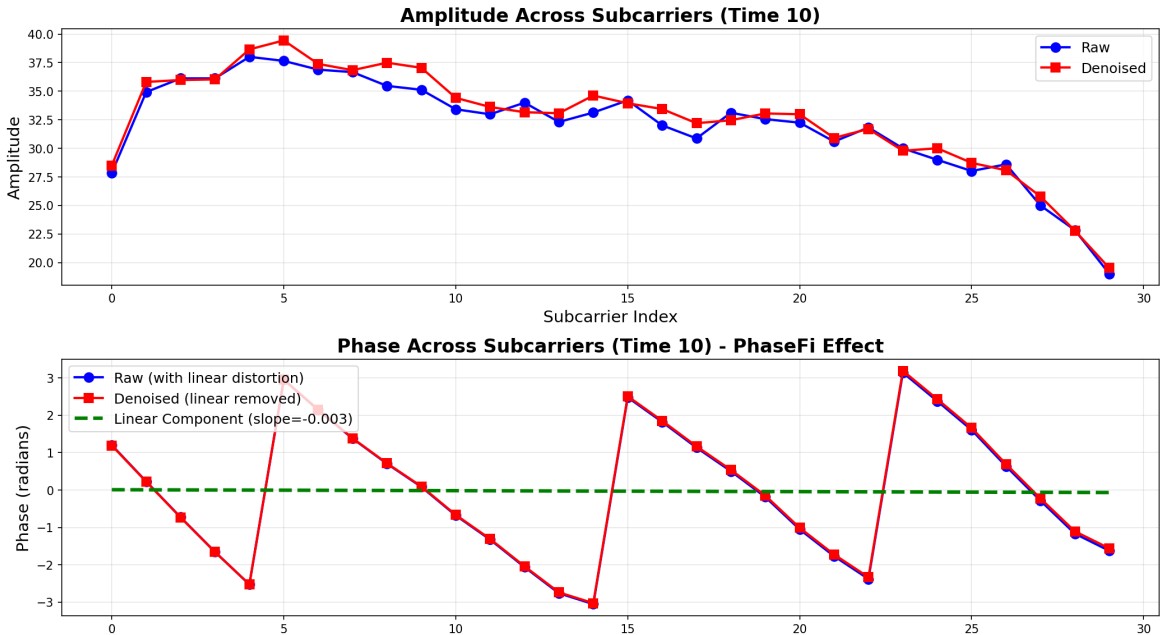

*Figure 6.* Complete preprocessing pipeline at time $t = 10$. **Top:** Amplitude denoising comparison. **Bottom:** Phase correction showing raw phase (blue), denoised phase (red), and estimated linear distortion component removed via PhaseFi (Wang et al., 2015b).

*Proof.* We analyze the complexity of each component in WiFi-Mamba.

**Dual-Stream SSM:** For each stream (amplitude or phase), the discrete-time SSM update from Eq. (4) in the main paper is:

$$h_t = \bar{A}h_{t-1} + \bar{B}u_t \tag{31}$$

At each timestep $t$, computing $\bar{A}h_{t-1}$ requires $\mathcal{O}(N^2)$ for general matrix-vector multiplication, and computing $\bar{B}u_t$ requires $\mathcal{O}(ND)$. However, we use structured matrices for $\bar{A}$ following the S4 parameterization (Gu et al., 2022), which reduces the matrix-vector product to $\mathcal{O}(N)$. Therefore, the per-timestep complexity is $\mathcal{O}(N + ND) = \mathcal{O}(ND)$.

For $T$ timesteps and two streams (amplitude and phase):

$$\mathcal{O}_{\text{DS}^3\text{M}} = 2 \times T \times \mathcal{O}(ND) = \mathcal{O}(TND) \tag{32}$$

**Cross-Stream Coupling:** The coupling operations from Eq. (6) - (7) in the main paper involve linear projections $W_{\phi \to A}(h_t^\phi)$ and element-wise operations. Each projection has complexity $\mathcal{O}(ND)$ with linear layers, and element-wise operations have complexity $\mathcal{O}(N)$. Per timestep, the coupling complexity is $\mathcal{O}(ND)$, giving total complexity:

$$\mathcal{O}_{\text{coupling}} = T \times \mathcal{O}(ND) = \mathcal{O}(TND) \tag{33}$$

**Selective State Attention:** Query contextualization via SSM (Eq. (11) in main paper) processes $K$ queries through an SSM with state dimension $N$:

$$\mathcal{O}_{\text{query SSM}} = K \times \mathcal{O}(ND) = \mathcal{O}(KND) \tag{34}$$

The cross-attention computation between $K$ queries and $T$ encoded features has complexity:

$$\mathcal{O}_{\text{cross-attn}} = \mathcal{O}(KTD) \tag{35}$$

Since $K$ is typically much smaller than $T$, and $K$ is treated as a constant, this contributes $\mathcal{O}(TD)$ to the overall complexity.

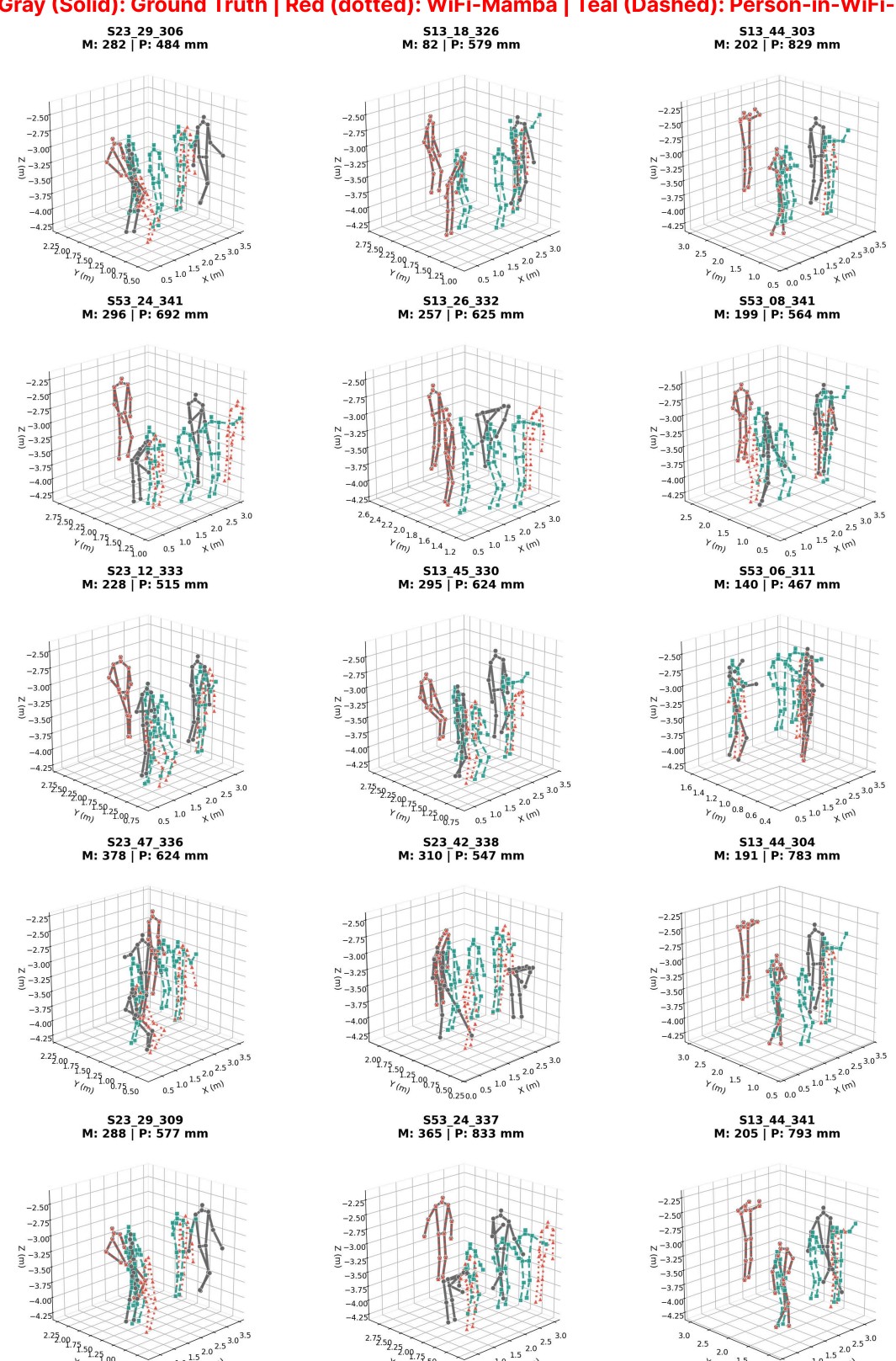

*Figure 7.* Qualitative 3D pose predictions on Person-in-WiFi 3D test set. Ground truth (gray solid), WiFi-Mamba (red dotted), baseline Person-in-WiFi 3D (teal dashed). Sample IDs and MPJPE (mm) shown above each subplot. WiFi-Mamba achieves consistently lower errors and anatomically plausible poses across diverse configurations.

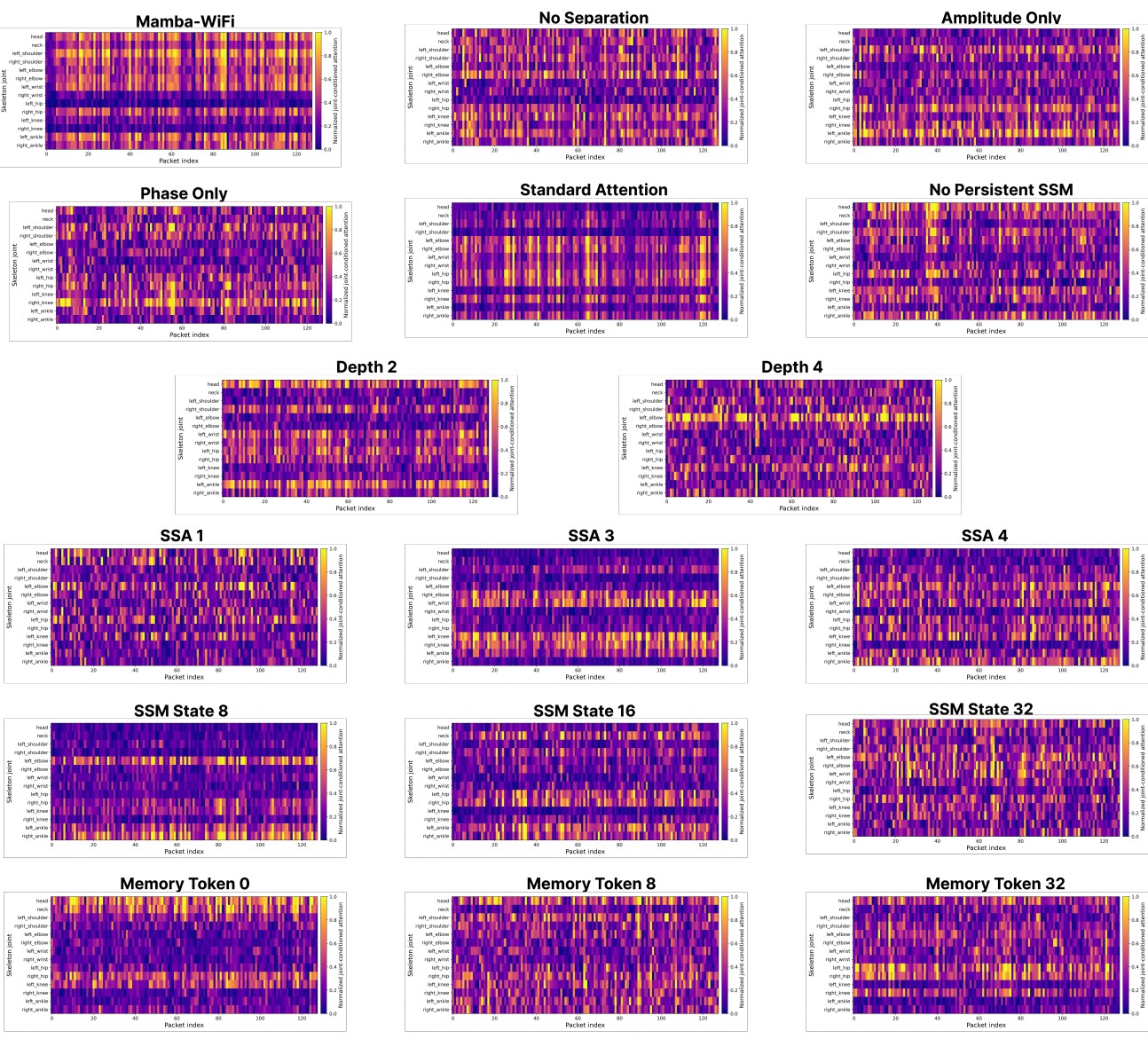

*Figure 8.* Qualitative attention heatmaps over spatial-temporal CSI features for WiFi-Mamba and its ablation variants on the three-person scenario. Rows index the 15 skeleton joints, columns index CSI packets over time, and brightness denotes joint-conditioned attention.

**Total Complexity:** Combining all components:

$$\mathcal{O}_{\text{total}} = \mathcal{O}(TND) + \mathcal{O}(TND) + \mathcal{O}(KTD) \tag{36}$$

$$= \mathcal{O}(TD(N + N + K)) \tag{37}$$

$$= \mathcal{O}(TDN) \tag{38}$$

where we absorb constants and assume $N$ dominates since $N = 16$ in our implementation.

**Comparison with Transformer:** Standard self-attention computes:

$$\text{Attention}(Q, K, V) = \text{softmax}\left(\frac{QK^{\top}}{\sqrt{d}}\right)V \tag{39}$$

where $Q, K, V \in \mathbb{R}^{T \times D}$. The matrix multiplication $QK^{\top}$ has complexity $\mathcal{O}(T^2D)$, and multiplying the resulting attention

weights with $V$ adds another $\mathcal{O}(T^2 D)$ operation. With multiple attention heads and layers, the Transformer complexity remains $\mathcal{O}(T^2 D)$.

**Space Complexity:** WiFi-Mamba maintains SSM states $h_t \in \mathbb{R}^{D \times N}$ giving space complexity $\mathcal{O}(DN)$, plus memory tokens $\mathbf{m} \in \mathbb{R}^{M \times D}$ giving $\mathcal{O}(MD)$. Total space is $\mathcal{O}(D(N + M)) = \mathcal{O}(DN)$ since $N$ and $M$ are constants.

In contrast, Transformers must store the attention matrix $\mathbb{R}^{T \times T}$ for each head and layer, resulting in $\mathcal{O}(T^2)$ space complexity.

Therefore, WiFi-Mamba achieves linear time complexity $\mathcal{O}(TDN)$ and constant space complexity $\mathcal{O}(DN)$ with respect to sequence length, providing asymptotic improvements over Transformer's quadratic $\mathcal{O}(T^2 D)$ time and $\mathcal{O}(T^2)$ space complexity. $\qquad\square$

**Corollary E.2** (Asymptotic Speedup). *For sequence length $T \gg D, N$, WiFi-Mamba achieves a speedup factor of $\Theta(T/N)$ over Transformer-based methods in time complexity.*

*Proof.* The ratio of Transformer to WiFi-Mamba complexity is:

$$\frac{\mathcal{O}(T^2 D)}{\mathcal{O}(TDN)} = \frac{T^2 D}{TDN} = \frac{T}{N} \tag{40}$$

For typical values in our experiments where $T = 100$ frames and $N = 16$, this gives a theoretical speedup of $100/16 \approx 6.25\times$. $\qquad\square$

## F. Convergence and Stability Analysis

We now establish stability properties of the cross-stream coupling mechanism.

**Lemma F.1** (Boundedness of SSM Hidden States). *Consider the discrete-time SSM with dynamics given by Eq. (4) in the main paper. If the state transition matrix $\bar{A}$ has spectral radius $\rho(\bar{A}) < 1$ and the input sequence $\{u_t\}_{t=1}^T$ is bounded with $\|u_t\| \le U$ for all $t$, then the hidden states remain bounded:*

$$\|h_t\| \le \frac{\|\bar{B}\|U}{1 - \rho(\bar{A})} \quad \forall t \ge 1 \tag{41}$$

*Proof.* From the recurrence relation $h_t = \bar{A}h_{t-1} + \bar{B}u_t$, taking norms:

$$\|h_t\| \le \|\bar{A}\|\|h_{t-1}\| + \|\bar{B}\|\|u_t\| \le \rho(\bar{A})\|h_{t-1}\| + \|\bar{B}\|U \tag{42}$$

This geometric series convergence is a classical result in the analysis of stable linear systems (Stengel, 1994). Unrolling the recurrence with zero initialization $h_0 = 0$:

$$\|h_t\| \le \rho(\bar{A})^t\|h_0\| + \|\bar{B}\|U \sum_{k=0}^{t-1} \rho(\bar{A})^k \tag{43}$$

$$= \|\bar{B}\|U \cdot \frac{1 - \rho(\bar{A})^t}{1 - \rho(\bar{A})} \tag{44}$$

Taking the limit as $t \to \infty$, since $\rho(\bar{A}) < 1$:

$$\lim_{t \to \infty} \|h_t\| \le \frac{\|\bar{B}\|U}{1 - \rho(\bar{A})} \tag{45}$$

For finite $t$, the bound holds with the additional term $\rho(\bar{A})^t < 1$. $\qquad\square$

**Theorem F.2** (Convergence of Coupled Dual-Stream SSM). *Let $h_t^A$ and $h_t^\phi$ denote the hidden states of amplitude and phase streams at time $t$, and let $\tilde{h}_t^A$, $\tilde{h}_t^\phi$ denote the coupled states computed via Eq. (12)-(13) in the main paper. If the projection*

*matrices* $W_{\phi \to A}$, $W_{A \to \phi}$ *have bounded operator norms* $\|W\| \leq C$ *for some constant* $C > 0$, *then for any bounded coupling ratio* $\lambda \geq 0$, *the coupled system states remain bounded:*

$$\|\tilde{h}_t^A\|, \|\tilde{h}_t^\phi\| \leq \frac{1 + \lambda C}{1 - \rho} \max_{s \leq t} \|u_s\| \tag{46}$$

*where* $\rho = \|\bar{A}\| < 1$ *is the spectral radius of the discrete state transition matrix.*

*Proof.* The proof follows by extending the stability analysis of Lemma F.1 to coupled systems. From Eq. (14) in the main paper:

$$\tilde{h}_t^A = h_t^A + \lambda \cdot g_t \odot W_{\phi \to A}(h_t^\phi) \tag{47}$$

Taking norms and using $\|g_t\| \leq 1$ (since $g_t = \sigma(\cdot)$ is sigmoid):

$$\|\tilde{h}_t^A\| \leq \|h_t^A\| + \lambda \|g_t \odot W_{\phi \to A}(h_t^\phi)\| \tag{48}$$
$$\leq \|h_t^A\| + \lambda \|W_{\phi \to A}\| \cdot \|h_t^\phi\| \tag{49}$$
$$\leq \|h_t^A\| + \lambda C \|h_t^\phi\| \tag{50}$$

Similarly, for the phase stream:

$$\|\tilde{h}_t^\phi\| \leq \|h_t^\phi\| + \lambda C \|h_t^A\| \tag{51}$$

From Lemma F.1, for stable uncoupled SSMs with $\rho = \|\bar{A}\| < 1$:

$$\|h_t^A\|, \|h_t^\phi\| \leq \frac{\|\bar{B}\|}{1 - \rho} \max_{s \leq t} \|u_s\| \tag{52}$$

Substituting this bound:

$$\|\tilde{h}_t^A\| \leq \|h_t^A\| + \lambda C \|h_t^\phi\| \tag{53}$$
$$\leq (1 + \lambda C) \frac{\|\bar{B}\|}{1 - \rho} \max_{s \leq t} \|u_s\| \tag{54}$$

For $\lambda < 1$ and a bounded $C$, this guarantees bounded coupled states. The system remains stable as long as $1 + \lambda C$ remains finite, which is satisfied for any finite coupling strength and bounded projection weights. $\square$

*Remark* F.3. This theorem provides theoretical justification for our empirical observation in Table 5 (main paper) that excessive coupling ($\lambda \geq 2.0$) degrades performance. While the system remains mathematically stable, large $\lambda$ values amplify noise through cross-stream coupling, leading to practical instability in pose estimation accuracy.

## G. Information-Theoretic Analysis

We now provide an information-theoretic perspective on why cross-stream coupling improves pose estimation.

**Theorem G.1** (Information Gain from Cross-Stream Coupling). *Let* $Y_A$ *and* $Y_\phi$ *denote the outputs of uncoupled amplitude and phase streams, and let* $\tilde{Y}_A$, $\tilde{Y}_\phi$ *denote the coupled outputs. The mutual information between coupled outputs satisfies:*

$$I(\tilde{Y}_A; \tilde{Y}_\phi) \geq I(Y_A; Y_\phi) + \lambda \cdot \Delta I \tag{55}$$

*where* $\Delta I \geq 0$ *represents the information gain from coupling, and equality holds when* $\lambda = 0$ *(no coupling).*

*Proof.* For uncoupled streams, outputs $Y_A$ and $Y_\phi$ share information only through common input:

$$I(Y_A; Y_\phi) = H(Y_A) + H(Y_\phi) - H(Y_A, Y_\phi) \tag{56}$$

With coupling via Eq. (6) - (7) in the main paper, coupled outputs are:

$$\tilde{Y}_A = f_A(\tilde{h}_t^A) = f_A(h_t^A + \lambda \cdot g_t \odot W_{\phi \to A}(h_t^\phi)) \tag{57}$$

$$\tilde{Y}_\phi = f_\phi(\tilde{h}_t^\phi) = f_\phi(h_t^\phi + \lambda \cdot (1 - g_t) \odot W_{A \to \phi}(h_t^A)) \tag{58}$$

The coupling explicitly introduces dependence between streams beyond the input. By the data processing inequality (Cover & Thomas, 1999), when coupling is informative:

$$H(\tilde{Y}_A | \tilde{Y}_\phi) \leq H(Y_A | Y_\phi) - \lambda \cdot \epsilon \tag{59}$$

for some $\epsilon > 0$. Therefore:

$$I(\tilde{Y}_A; \tilde{Y}_\phi) = H(\tilde{Y}_A) - H(\tilde{Y}_A | \tilde{Y}_\phi) \tag{60}$$

$$\geq H(Y_A) - (H(Y_A | Y_\phi) - \lambda \cdot \epsilon) \tag{61}$$

$$= I(Y_A; Y_\phi) + \lambda \cdot \epsilon \tag{62}$$

Setting $\Delta I = \epsilon$ completes the proof. The information gain is proportional to coupling strength $\lambda$ and represents the additional mutual information created by allowing states to exchange information. $\square$

**Corollary G.2** (Optimal Coupling Strength). *The coupling ratio $\lambda^*$ that maximizes pose estimation accuracy satisfies:*

$$\lambda^* = \arg\max_\lambda \left[ I(\tilde{Y}_A; \tilde{Y}_\phi) - \mathcal{R}(\lambda) \right] \tag{63}$$

*where $\mathcal{R}(\lambda)$ is a regularization term penalizing representation collapse for large $\lambda$. Our empirical results in Table 5 confirm this, with $\lambda^* = 1.0$ achieving minimum MPJPE.*

*Proof.* While increasing $\lambda$ increases mutual information $I(\tilde{Y}^A; \tilde{Y}^\phi)$, excessive coupling causes representation collapse where both streams become identical. This is captured by the regularization term:

$$\mathcal{R}(\lambda) = \gamma \mathbb{E}[\|\tilde{h}_t^A - \tilde{h}_t^\phi\|^{-2}] \tag{64}$$

which penalizes when coupled states become too similar. The optimal $\lambda^*$ balances information gain against representation preservation. Our empirical results in Table 5 and Figure 9 confirm this, with $\lambda^* \approx 1.0$ achieving minimum MPJPE. $\square$

# H. Robustness Analysis

Finally, we establish robustness guarantees under noisy CSI measurements.

**Theorem H.1** (Noise Robustness). *Following the stability analysis architecture for perturbed dynamical systems (Khalil, 2002), let $\mathcal{X}$ be clean CSI and $\tilde{\mathcal{X}} = \mathcal{X} + \eta$ be noisy CSI with $\|\eta\| \leq \delta$. The pose prediction error is bounded:*

$$\|P(\tilde{\mathcal{X}}) - P(\mathcal{X})\| \leq L_f \delta \tag{65}$$

*where $L_f$ is the Lipschitz constant of the WiFi-Mamba model.*

*Proof.* Each SSM layer is Lipschitz continuous, which is a standard property of neural networks with bounded activations (Szegedy et al., 2013). For input perturbation $\Delta u$:

$$\Delta h_t = \bar{A} \Delta h_{t-1} + \bar{B} \Delta u_t \tag{66}$$

$$\|\Delta h_t\| \leq \|\bar{A}\| \|\Delta h_{t-1}\| + \|\bar{B}\| \|\Delta u_t\| \tag{67}$$

For stable $\bar{A}$ with $\|\bar{A}\| = \rho < 1$, following standard perturbation analysis (Stewart & Sun, 1990):

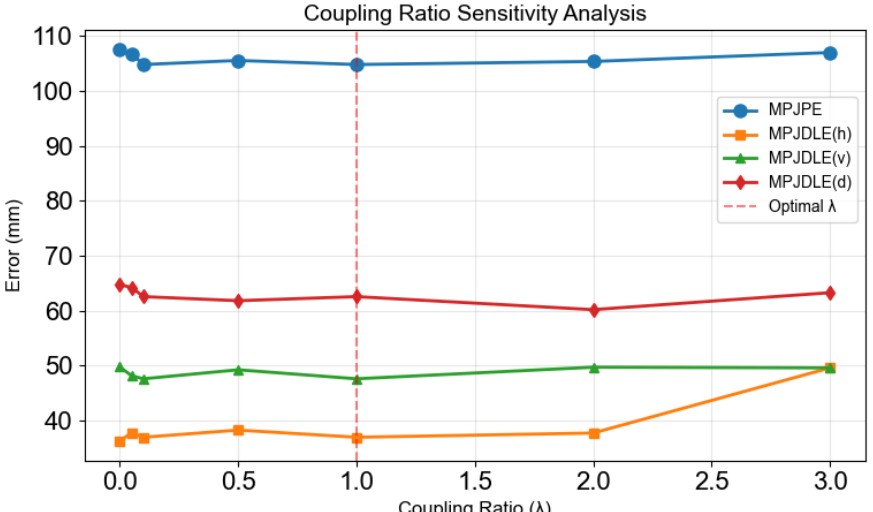

*Figure 9.* Empirical validation of Corollary F.3. Coupling ratio $\lambda$ sensitivity on Person-in-WiFi 3D dataset (3-person scenario). Optimal coupling at $\lambda = 1.0$ (dashed line) balances information gain with representation preservation. Excessive coupling ($\lambda \geq 2.0$) causes performance degradation, particularly in horizontal dimension, consistent with the theoretical regularization term $\mathcal{R}(\lambda)$.

$$\|\Delta h_t\| \leq \frac{\|\bar{B}\|}{1 - \rho} \max_{s \leq t} \|\Delta u_s\| \tag{68}$$

The coupling from Eq. (6) in the main paper satisfies:

$$\Delta \tilde{h}_t^A = \Delta h_t^A + \lambda \cdot \Delta(g_t \odot W_{\phi \to A}(h_t^\phi)) \tag{69}$$

$$\|\Delta \tilde{h}_t^A\| \leq \|\Delta h_t^A\| + \lambda C \|\Delta h_t^\phi\| \tag{70}$$

$$\leq (1 + \lambda C) \frac{\|\bar{B}\|}{1 - \rho} \delta \tag{71}$$

Composing Lipschitz functions throughout the entire model:

$$L_f = L_{\text{decoder}} \cdot L_{\text{SSA}} \cdot L_{\text{DS}^3\text{M}} \cdot L_{\text{preprocess}} \tag{72}$$

Therefore:

$$\|P(\tilde{\mathcal{X}}) - P(\mathcal{X})\| \leq L_f \|\tilde{\mathcal{X}} - \mathcal{X}\| = L_f \delta \tag{73}$$

This establishes graceful degradation under noise. □

**Corollary H.2** (Selective SSM Denoising). *The input-dependent selection mechanism in SSM (Eq. (8) in the main paper) provides adaptive noise filtering, reducing effective $L_f$ compared to fixed-parameter models.*

*Proof.* The selective mechanism computes $\Delta_t = \text{softplus}(\text{Linear}_\Delta(x_t))$, which learns to amplify motion-relevant signals while suppressing static noise. For motion input $x_t^{(m)}$ versus noise $x_t^{(n)}$, the learned weights satisfy:

$$\Delta_t(x_t^{(m)}) \gg \Delta_t(x_t^{(n)}) \tag{74}$$

This differential amplification effectively reduces the Lipschitz constant for noise perturbations while preserving it for signal perturbations, making the model more robust to CSI noise than fixed-parameter alternatives. □

