# OpenReview forum: "Interleaved Selective State Space Models for Efficient WiFi-Based 3D Multi-Person Pose Estimation"
_ICML.cc/2026/Conference — ICML 2026 regular_

### Official Review · Reviewer_Vs8T · 2026-03-04

**Soundness:** 3
**Presentation:** 3
**Significance:** 2
**Originality:** 3
**Overall Recommendation:** 4
**Confidence:** 2

**Summary:**

This paper proposes WiFi-Mamba, the first State Space Model (SSM) architecture tailored for WiFi-based 3D multi-person pose estimation. The authors introduce three key technical contributions: (1) a Dual-Stream Selective State Space Model (DS3M) that processes amplitude and phase through parallel pathways with cross-stream state coupling; (2) Selective State Attention (SSA) for pose query decoding with SSM-derived sequential context; and (3) Persistent SSM Memory for maintaining temporal consistency across frames. Extensive experiments on the Person-in-WiFi 3D dataset demonstrate 16-27% MPJPE reduction while using only 4.4% of baseline parameters (2.14M vs. 48.2M), suggesting promising efficiency-accuracy trade-offs for edge deployment in privacy-sensitive monitoring scenarios.

**Compliance With Llm Reviewing Policy:**

Affirmed.

**Key Questions For Authors:**

1.Have you considered validating WiFi-Mamba on other public WiFi-CSI datasets (e.g., WiPose) or cross-environment settings? If not, what adaptation strategies might mitigate domain shift in real-world deployment?
2.Could you supplement the ablation results with qualitative visualizations (e.g., 3D pose comparisons, attention heatmaps)? Such visual evidence would help readers intuitively understand how components like cross-stream coupling concretely improve estimation.
3.Could you discuss or provide preliminary results on WiFi-Mamba's performance under realistic perturbations (e.g., multipath interference, non-line-of-sight conditions)? Even a small-scale pilot study or sensitivity analysis would strengthen the argument for practical edge deployment.

**Limitations:**

yes

**Strengths And Weaknesses:**

Strengths
1.The motivation is well-grounded in real-world needs: aging population monitoring, privacy-preserving sensing, and occlusion-robust applications. The positioning of WiFi-based HPE as a complementary alternative to camera-based systems is compelling.
2.The paper is well-organized with a clear methodology section.
3.The authors propose an architecture that applies State Space Models (SSMs) to WiFi-based pose estimation, demonstrating a certain degree of originality.

Weaknesses
1. Evaluation relies solely on the Person-in-WiFi 3D dataset (3 indoor scenes, 9 subjects, fixed 3×3 MIMO configuration), leaving the method's generalization capability across diverse hardware platforms, channel conditions, and crowd densities unverified; validating on other public WiFi-CSI datasets (e.g., WiPose) or discussing cross-domain adaptation strategies would significantly strengthen the credibility of the conclusions.

2.Tables 3-5 provide solid quantitative metrics, but numerical comparisons alone cannot intuitively illustrate how components like cross-stream coupling or persistent memory improve pose estimation in challenging scenarios; supplementing with qualitative visualizations (e.g., 3D pose prediction comparisons, attention heatmaps, or state evolution trajectories) would help readers more intuitively understand the method's advantages and enhance the interpretability of technical insights.

3. Experiments are primarily conducted in controlled laboratory settings, yet the paper targets "privacy-sensitive edge deployment," and real-world factors (e.g., multipath interference, device heterogeneity, non-line-of-sight propagation) are not systematically evaluated; including pilot studies in real-world environments or sensitivity analysis under realistic perturbations would more comprehensively demonstrate the method's practical boundaries and deployment feasibility given that the reported MPJPE of 76.75mm may approach usability thresholds for applications like fall detection.

---

> ### Author Rebuttal · Authors · 2026-03-31
>
> **Q1: Model generalization across dataset**
>
> **A:** Please refer to our response to Reviewer Vxfi (Q2) for further details.
>
> **Q2: Supplement the ablation results with qualitative visualizations**
>
> **A:** Thank you for this suggestion. To complement the ablation study, we have plotted the qualitative heatmap visualizations showing the model attention over spatial-temporal CSI features (See: https://figshare.com/s/b5911f78bad970e5c881?file=63337840). These visualizations demonstrate that our proposed design produces more structured, joint-consistent activation patterns than baseline variants, indicating improved feature representation and robustness. The figure will be included in the revised version.
>
> **Q3: Model sensitivity analysis under perturbation and application on fall detection**
>
> **A:** We thank the reviewer for this suggestion. In response, we have added a model sensitivity analysis that simulates key real-world factors, including multipath interference and NLOS propagation. We perturb the CSI tensor, with multipath simulated via frequency-selective fading and delayed reflected paths, while NLOS is modeled through amplitude attenuation, per-antenna variation, and SNR-controlled phase and amplitude noise. The performance is shown in Table 1 and Table 2 showing the stabilities of the architecture under perturbation, we will include these results in the revised version of the paper.
>
> ### Table A: Sensitivity analysis under simulated multipath interference
>
> | Condition | Paths | Faded (%) | MPJPE | PA-MPJPE | MPJPE (h) | MPJPE (v) | MPJPE (d) | PCK@30 |
> |----------|-------|-----------|--------|-----------|------------|------------|------------|---------|
> | Clean    | 0     | 0         | 101.36 | 60.72     | 35.91      | 48.24      | 58.14      | 75.9    |
> | Mild     | 1     | 3         | 111.50 | 64.35     | 37.64      | 52.50      | 65.77      | 72.8    |
> | Moderate | 2     | 7         | 167.32 | 73.92     | 75.40      | 68.57      | 82.66      | 60.5    |
> | Severe   | 3     | 12        | 228.09 | 84.14     | 118.24     | 89.09      | 103.11     | 48.5    |
> | Extreme  | 5     | 20        | 294.05 | 99.71     | 166.36     | 111.20     | 125.98     | 36.3    |
>
> ### Table B: Sensitivity analysis under simulated NLOS conditions
>
> | Condition       | Reduction (%) | Phase SNR | MPJPE | PA-MPJPE | MPJPE (h) | MPJPE (v) | MPJPE (d) | PCK@30 |
> |----------------|---------------|-----------|--------|-----------|------------|------------|------------|---------|
> | LOS (Clean)    | 0             | $\infty$         | 101.36 | 60.72     | 35.91      | 48.24      | 58.14      | 75.9    |
> | Partial NLOS   | 10            | 30        | 100.67 | 60.44     | 35.54      | 48.13      | 57.76      | 75.3    |
> | Moderate NLOS  | 25            | 20        | 105.72 | 61.52     | 36.04      | 51.87      | 60.97      | 72.9    |
> | Full NLOS      | 45            | 12        | 251.89 | 81.11     | 137.91     | 93.39      | 105.72     | 46.2    |
> | Extreme NLOS   | 70            | 6         | 625.48 | 130.91    | 373.02     | 184.16     | 282.35     | 19.0    |
>
> Regarding the reported 76.75mm MPJPE, we agree that this level of accuracy approaches the usability threshold for applications such as fall detection, which would better characterize the method’s practicality. This was also one of our initial intentions, but due to the lack of a hardware setup required for deployment, we are currently unable to conduct experiments for this work. In addition, the current dataset does not include fall actions, which further limits evaluation in this direction. We acknowledge these as limitations and will discuss them explicitly in future work.

---

> > ### Author Rebuttal · Reviewer_Vs8T · 2026-04-03
> >
> > Thank you to the authors for providing both qualitative and quantitative experiments, which have addressed my concerns.

---

> > > ### Author Response · Authors · 2026-04-05
> > >
> > > We appreciate the reviewer's time and positive feedback. We're glad our newly included experimental results and analysis address your concerns.

---

### Official Review · Reviewer_eFmm · 2026-03-06

**Soundness:** 3
**Presentation:** 2
**Significance:** 2
**Originality:** 4
**Overall Recommendation:** 4
**Confidence:** 4

**Summary:**

This paper tackles the problem of 3D multi-person pose estimation using WiFi Channel State Information (CSI) signals. The authors point out that current state-of-the-art methods rely heavily on Transformers, which means they suffer from quadratic computational complexity and end up being massive parameter-wise. Plus, these generic architectures don't really take advantage of the underlying physics of WiFi signals.To solve this, the paper introduces "WiFi-Mamba," an architecture built on State Space Models (SSMs) to achieve linear complexity. The model is tested on the Person-in-WiFi 3D dataset. The model outperforms the heavy Transformer baseline, dropping the error (MPJPE) by about 16% to 27% across different multi-person scenarios. More importantly, it does this while shrinking the model size from 48.2M down to just 2.14M parameters, making it much more viable for real-time edge deployment.

**Compliance With Llm Reviewing Policy:**

Affirmed.

**Ethical Review Concerns:**

None.

**Final Justification:**

I am raising my score to a Weak Accept after reading the rebuttal. The author addressed my primary concerns regarding the paper's soundness and presentation. The video results, revised figures and experiments provided the necessary empirical support. The explanation of the statistical analysis regarding amplitude and phase, combined with a discussion of the dual-process system, improved the paper's clarity and made the underlying mechanisms easier to understand.

**Key Questions For Authors:**

1. Regarding the motivation behind your technical contributions. While the motivation for the overall task is explained thoroughly in the introduction, the deeper insights behind your specific technical choices seem to be missing. Your ablation studies definitely prove that the method is effective, but understanding exactly why it works is just as important. For instance, you mentioned that the model size is significantly reduced compared to Transformer-based methods, but why does this heavily reduced model actually outperform the much more complicated baseline? Also, why exactly is the proposed dual-stream system more suitable for processing CSI signals? You make the claim that amplitude captures signal strength variations due to body movements and occlusions, while phase encodes fine-grained Doppler shifts induced by velocity. This is a strong physical claim that really needs further theoretical or empirical justification. I have a similar question about the Selective State Attention module. You mentioned that queries representing body parts should be contextually aware, but this intuition probably needs further experimental justification to show exactly how that awareness translates to the performance gains. Ideally, these kinds of core insights should be highlighted right there in the introduction section.
2. My other major question is about the qualitative results, which I touched on earlier. I would really like to hear your explanation on why the poses in the figures are so hard to visually recognize, including the ground truth ones. Also, why do some of these qualitative visualizations seem to misalign with your strong quantitative results? Seeing the baseline visually outperform the proposed method in certain frames makes it a bit confusing to appreciate the improvements you reported.

These are my two major questions, and I am very open to raising my rating of the paper if these points can be thoroughly addressed in the rebuttal.

**Limitations:**

Yes. I agree with the authors that the main limitation for this entire task is real-world deployment. Until we can get this running reliably in the wild, the advantages it has over standard vision-based pose estimation methods just aren't that significant. I would really like to hear the authors' thoughts on how we might actually bridge this gap moving forward.

**Strengths And Weaknesses:**

### Soundness

The paper presents an SSM-based algorithm for WiFi-based human pose estimation. I think the main technical contribution really lies in the dual-stream SSM architecture and the selective state attention used for pose queries. The technical method is explained thoroughly and the details are included in the appendix.

### Presentation

The writing is quite clear, but I do have a few suggestions. The figures, in my opinion, are a bit hard to follow right now.

1. I guess the intention for Figure 1 (the teaser figure) is to highlight the overall contribution, but I found it a bit too information-dense. The authors are trying to pack the dual-stream architecture, the SSA module, the task definition (input/output), and the results all into one small figure. It actually took me reading the entire paper to fully understand it, so I think this could definitely use some tweaking. For instance (just as a suggestion), only show the task input/output and some qualitative results there, skipping the technical details. This way, even readers who aren't experts in WiFi-based HPE (like me) can get a quick grasp of what the paper is about and how good the results are. Also, the amplitude and phase plots in Figure 1 might not be the most intuitive way to show the data. This brings me to Figure 2: I think the dual-stream pipeline could use a more straightforward visual representation. Right now, the whole thing looks like a pile of similarly shaped blocks, which takes some time to digest. Also, it would really help the readers if the authors cited Figure 2 throughout Section 3 as explaining the method, rather than just mentioning it once.
2. Minor issue: the fonts in Figure 3 are inconsistent. The text sizes on the left and right sides do not match. Also, some of the fonts are just too small and should be adjusted to match the size of the main text.
3. One more thing. I wanted to point out that for a vision/graphics task like human pose estimation, the qualitative results here are a bit hard to read. The authors provided these in Figure 7, but my question is: why are some of the poses so hard to make out (even the gray solid ground truth ones)? For some of them, I can somewhat tell it looks like a person, but for others, I can barely recognize a human pose at all. Also, the advantage of the proposed method is not super obvious in these qualitative results. For example, in the 4th row, 2nd column, the red dotted Mamba prediction is out of the frame, while the green baseline actually looks closer to the GT. Why is this? This is quite an important point for me.

### Significance

I think the task itself is quite relevant, and in my opinion, the paper provides a novel and effective solution for WiFi-based human pose estimation. Since you claim that a major advantage of your solution is the heavily reduced model size, it would be really nice to see some video demos of live pose estimation running. Also, I have a few questions about the motivation behind the specific technical contributions, but I will elaborate on those in the questions section.

### Originality

The paper introduces a novel method for the WiFi-based HPE task. I have no questions here.

---

> ### Author Rebuttal · Authors · 2026-03-31
>
> **Q1: Figure 1 is too dense and not intuitive, Figure 2 is visually repetitive and under-referenced in Section 3, and Figure 3 has inconsistent fonts.**
>
> **A:** Following your suggestion, we have redesigned Figure 1, and Figure 2 for clarity. We will reference Figure 2 throughout Section 3 to better guide the methodology, and we will standardize the font sizes in Figure 3 to match the main text in the revision. (See: https://figshare.com/s/73575b26ddbead38d211 for the revised Figures 1, 2, and 3)
>
> **Q2: Video demo pose estimation inference.**
>
> **A:** Thanks for this suggestion. Due to the lack of equipment required for deployment, we are unable to provide a video demo for this work. However, we will consider it in our future work. For now, we can provide a video demo of the prediction generated from the data sequence. (See: https://figshare.com/s/7fad7749a9144709a707 for demo video)
>
> **Q3: Why does the proposed model, which is a heavily reduced model, actually outperform the much more complicated baseline? Also, why is the proposed dual-stream system more suitable for processing CSI signals?**
>
> **A:** Regarding why the reduced model outperforms the larger baseline, we attribute it to the baseline architecture being unnecessarily dense for the WiFi-CSI HPE task, resulting in computational overhead without proportional accuracy gains. This is consistent with trends in the field, where recent works have shown competitive performance with significantly fewer parameters, including MetaFi++ (26.5M), DT-Pose (4.8M), and HPE-Li (0.8M).
>
> And we attribute this to the inductive biases of SSMs being better suited to sequential WiFi-CSI signals than Transformers. Transformers treat all positions equally via global attention, while SSMs naturally encode temporal ordering through recurrent state propagation, which aligns with the causal structure of CSI.
>
> **Q4: Evidence supporting the claim that amplitude captures signal strength variations while phase encodes Doppler-induced motion information?**
>
> **A:** We acknowledge that claiming amplitude captures body shadowing and phase encodes Doppler shifts is an oversimplification. From the CSI propagation model in Eq. (1), both amplitude and phase are functions of the same underlying parameters ${\alpha_k(t), \tau_k(t)}$ and are therefore not strictly physically separable.
>
> A more rigorous motivation lies in the statistical symmetry between the two modalities. Amplitude is a strictly non-negative quantity with a smooth, skewed distribution shaped by multipath superposition [1]. Phase, in contrast, is a signed periodic quantity defined over $[-\pi, \pi]$, and is highly susceptible to hardware-induced carrier frequency offset (CFO) and sampling frequency offset (SFO), which introduce phase offsets and linear rotation errors across subcarriers [2]. These statistical differences make shared parameterization difficult, motivating separate processing streams with independent parameters. Also, Table 3 results have confirmed their complementarity. We will revise the manuscripts to replace the physical claim with this statistical argument.
>
> **Q5: What experiments show that the Selective State Attention actually improves performance, and why does it help?**
>
> **A:** Regarding SSA, the contextual awareness across queries is evidenced by the 13.1% MPJPE degradation when replacing SSA with standard cross-attention in Table 3, suggesting that inter-query context is critical for resolving multi-person ambiguity in shared CSI observations. We will bring these insights into the introduction in the revision.
>
> **Q6: Qualitative visualization misalignment.**
>
> **A:** We thank the reviewer for this comment. We have re-run the qualitative visualization for multi-human pose estimation and will include the updated figure in the revision (See: https://figshare.com/s/6bf43e98b33407dfaeb4 for revised Figure 7).
>
> **Q7: Limitations and Gaps before deploying in the wild.**
>
> **A:** Cross-environment generalization remains largely unsolved, with only one existing work addressing it through domain adaptation [3], underscoring its underexplored despite its importance for real-world deployment. More diverse datasets covering a wider range of people, environments, and activities are essential for generalization beyond controlled lab conditions, even though dataset availability remains limited, with Person-in-WiFi 3D as the only publicly available dataset for multi-person WiFi-CSI-based pose estimation. And to our knowledge, there is no dataset that exists for a real-world environment.
>
> ### References:
> [1] Tse & Viswanath, Fundamentals of Wireless Communication, 2005.
>
> [2] Zhuo et al., Perceiving accurate CSI phases with commodity WiFi devices, INFOCOM, 2017.
>
> [3] Zhou et al., AdaPose: Toward cross-site device-free human pose estimation with commodity WiFi., IEEE IoT Journal, 2024.

---

> > ### Author Rebuttal · Reviewer_eFmm · 2026-04-02
> >
> > Thank you for the video results, revised figures and experiments. I appreciate the authors' explanation of the statistical analysis regarding amplitude and phase. The revised figures and experimental results, combined with a deeper discussion of the dual-process system, improve the paper's overall presentation and make the underlying mechanisms easier to understand. In light of these improvements, I am willing to raise my score to a Weak Accept.

---

> > > ### Author Response · Authors · 2026-04-02
> > >
> > > Thank you for your encouraging feedback and for noticing the improvements in our revision. We appreciate that the added video results, revised figures, and expanded experimental analysis have clarified the roles of amplitude and phase, along with demonstrating the effectiveness of the dual-process design.
> > >
> > > We especially value your recognition of the statistical analysis and the in-depth discussion of the system’s fundamental mechanisms, which were important elements we aimed to enhance during revision.
> > >
> > > Thank you again for your constructive comments and for reconsidering your evaluation.

---

### Official Review · Reviewer_UCSG · 2026-03-12

**Soundness:** 3
**Presentation:** 4
**Significance:** 3
**Originality:** 4
**Overall Recommendation:** 5
**Confidence:** 4

**Summary:**

This paper proposes a novel SSM-based architecture for WiFi-based pose estimation. The model is built on a dual-stream backbone that processes amplitude and phase separately with cross-stream coupling. Experiments on the Person-in-WiFi 3D dataset report new SOTA performance over existing WiFi-based methods while using much fewer parameters.

**Compliance With Llm Reviewing Policy:**

Affirmed.

**Final Justification:**

The rebuttal has addressed my main concerns, and I will keep my rating.

**Key Questions For Authors:**

1. In Page 5, Eq. (17), what is the exact shape of the state-derived bias (b), and how is it added to the attention logits in the implementation?

**Limitations:**

Yes.

**Strengths And Weaknesses:**

Strengths:
1. The paper establishes a new efficiency-accuracy trade-off and achieves MPJPE improvements over baselines significantly while reducing the parameters. WiFi-Mamba improves MPJPE from 91.77 to 76.75 against Person-in-WiFi 3D while reducing parameters from 48.2M to 2.14M.

2. The high-level design is well motivated. Separating amplitude and phase is reasonable. The explanation is very thorough, including clear figures and detailed formulations.

3. The paper includes nontrivial ablations which analysis dual-stream modeling, persistent memory, and the proposed attention module.

Weaknesses:
1. The individual component lacks novelty. The novelty lies in their combination and adaptation to WiFi-CSI, which is legitimate but may be argued more explicitly.

2. On Page 5, Eq. (17), the attention bias (b) is defined as a vector in (\mathbb{R}^H), one scalar per head. If that scalar is broadcast over all key positions for a head, it adds a constant to every logit and therefore disappears under softmax.

---

> ### Author Rebuttal · Authors · 2026-03-31
>
> **Q1: The individual component lacks novelty. The novelty lies in their combination and adaptation to WiFi-CSI, which is legitimate but may be argued more explicitly.**
>
> **A:** We agree that the individual components draw from existing techniques, but we respectfully note that the novelty of our work lies in their principled combination. More importantly, their adaptation to the unique characteristics of WiFi-CSI signals. In our revision, we will explicitly articulate this by highlighting the physics-informed motivation for adapting each component to CSI’s amplitude-phase structure, and emphasizing how their synergistic integration addresses the distinct challenges of multi-person WiFi-based pose estimation that prior methods have not explored.
>
> **Q2: On Page 5, Eq. (17), the attention bias $b \in \mathbb{R}^H$, one scalar per head. If that scalar is broadcast over all key positions, it adds a constant and disappears under softmax. And how is it added to the attention logits in the implementation?**
>
> **A:** Thanks for this observation. We agree that, as written, $b \in \mathbb{R}^H$ is broadcast uniformly across all key positions for each head and therefore cancels under the softmax:
> $$
> \text{softmax}\left(\frac{\tilde{Q}K^\top}{\sqrt{d}} +
> \tau \cdot b\right) = \text{softmax}\left(
> \frac{\tilde{Q}K^\top}{\sqrt{d}}\right)
> $$
> since adding a constant to all logits does not affect the softmax distribution. In the implementation, $b$ is reshaped
> to $(B, H, 1, 1)$ and broadcast uniformly over all $(K \times N)$ logit positions per head, consistent with
> this cancellation. This bias term, therefore, does not contribute to the attention weights regardless of the value
> of $\tau$, and we will remove it in the revised manuscript. Importantly, the performance gain of SSA does not stem from this bias term, but from the SSM-based query transformation:
> $$
> \tilde{Q},; h^Q = \mathrm{SSM}(Q), \quad \tilde{Q} \in \mathbb{R}^{K \times D}.
> $$
> Here, the SSM processes queries sequentially, so each contextualized query $\tilde{Q}_k$ encodes information from preceding queries via the hidden state $h^Q$. This introduces inter-query dependency, in contrast to standard cross-attention, where queries are independent.
>
> Thereby, the attention logits $\tilde{Q}K^\top$ vary meaningfully across query positions, giving coherent reasoning over shared CSI observations. This is particularly important for multi-person pose estimation, where queries corresponding to different body parts or individuals must be jointly interpreted. We will also clarify this in the revised text to avoid confusion.

---

> > ### Author Rebuttal · Reviewer_UCSG · 2026-04-03
> >
> > Thanks for the rebuttal. The authors have addressed my concerns.

---

> > > ### Author Response · Authors · 2026-04-05
> > >
> > > We appreciate the reviewer's time and positive feedback. We're glad our response addressed your concerns.

---

### Official Review · Reviewer_Vxfi · 2026-03-13

**Soundness:** 3
**Presentation:** 3
**Significance:** 3
**Originality:** 3
**Overall Recommendation:** 4
**Confidence:** 3

**Summary:**

The paper proposes WiFi-Mamba for 3D multi-person pose estimation leveraging state space models. Experiments have been carried on Person-in-WiFi3Ddataset for both single and multi-person scenarios and demonstrate the efficiency.

**Compliance With Llm Reviewing Policy:**

Affirmed.

**Final Justification:**

The rebuttal has addressed my main concerns. It seems that the proposed method is more effective on multi-person pose estimation compared to single-person scenarios. Table B and C, there are some minor mark errors (bold number) for the best performance. For example, on Table B for column "PA-MPJPE", the best performance should be 60.09; Table C, Person-in-WiFi 3D for the first person performance, 94.96 and 86.20 should be bold. This is minor issues. It is hard to address the scalability issue due to the limitation of suitable dataset.

I will keep my rating.

**Key Questions For Authors:**

1. The introduction does not mention the challenges of WIFI-based 3D multi-person pose estimation compared single person. It is not clear that it is targeting solving multi-person scenarios.
2. Due to the limited dataset which hinder the comprehensive evaluation of the proposed methods. It is hard to know the generalization and scalability of the proposed method.
3. Multi-person dataset is rare. it would be nice to use the dataset in other single-person scenarios to compared the proposed method compared with other methods and include other evaluation metrics (e.g., PA-MPJPE, PCK) for comprehensive evaluation.

**Limitations:**

Partially discussed the limitations.

**Strengths And Weaknesses:**

1. The paper is technically sound and methods are used appropriate. The experiment design is fine, and evaluation are performed for single and multi-person scenarios, ablation studies have been formed to identify the effect of different components. However, the current evaluation is performed only in one dataset as we know that dataset will greatly affect the performance of the proposed methods. Meanwhile, the number of person is too less, it is hard to know the scalability of the proposed method. Meanwhile, it would be nice to include other evaluation metrics such as PA-MPJPE and PCK for comprehensive evaluation. Furthermore, it would be nice to identify why the proposed methods work better than transformer-based related work.
2. The paper is presentation in a very logical and clear way. Related works and difference are discussed. However, it would be nice to identify the challenges of WiFi-Based 3D multi-person pose estimation compared with single person.
3. The paper addresses the practical problem for 3D multi-person pose estimation leveraging Wifi and machine learning models. The proposed methods could be used in other applications. The current issue is that it is not sure the generalization of the proposed method as only evaluated on one dataset which is an issue.
4. The paper proposes WiFi-Mamba for 3D multi-person pose estimation using state space models to processes amplitude and phase in parallel. A dual-steam mechanism and selective state attention are proposed to enhance the cross coupling and attention. This is a good try to use state space models in WiFi-based 3D multi-person pose estimation.

---

> ### Author Rebuttal · Authors · 2026-03-31
>
> **Q1: The introduction does not mention the challenges of Wi-Fi-based 3D multi-person pose estimation compared to single-person. It is not clear that it is targeting solving multi-person scenarios.**
>
> **A:** Thank you for this remark. We will revise the introduction to clearly articulate the distinct challenges of multi-person WiFi-based pose estimation and clarify that our work is designed to address this scenario.
>
> **Q2: Due to the limited dataset, which hinders the comprehensive evaluation of the proposed methods. It is hard to know the generalization and scalability of the proposed method.**
>
> **A:** We acknowledge that the limited datasets constrain the comprehensiveness of the evaluation. Hence, to show the generalization and scalability of our proposed method, we have conducted an additional experiment on the suggested WiPose dataset. The model achieved competitive performance on the WiPose dataset as outlined in Table A. WiFi-Mamba is more parameter-efficient than the state-of-the-art, while the performance gap remains within only 1–2 points across most metrics, and the result is consistent with the Person-in-WiFi 3D dataset, and we will include it in the revised manuscript.
>
> ### Table A: Single-person pose estimation on WiPose dataset.
>
> | Method | Params (M) | MPJPE | PA-MPJPE | MPJPE (h) | MPJPE (v) | PCK@30 | PCK@20 | PCK@10 |
> |--------|------------|--------|-----------|------------|------------|---------|---------|---------|
> | MetaFi++ | 26.5 | 58.17 | 35.15 | 32.45 | 39.21 | 53.38 | 36.65 | 13.62 |
> | DT-Pose | 4.8 | **34.14** | **23.19** | **18.88** | **22.34** | **77.97** | **69.61** | _51.40_ |
> | HPE-Li | **0.8** | 39.83 | 25.52 | 22.38 | 26.58 | 70.58 | 56.87 | 33.21 |
> | **WiFi-Mamba (Ours)** | _2.14_ | _35.82_ | _24.06_ | _19.66_ | _23.62_ | _75.92_ | _67.75_ | **51.99** |
>
> **Q3: Multi-person dataset is rare. It would be nice to use the dataset in other single-person scenarios to compare the proposed method with other methods and include other evaluation metrics (e.g., PA-MPJPE, PCK) for comprehensive evaluation.**
>
> **A:** Thank you for this remark. We agree that multi-person datasets are relatively scarce, particularly for the specific problem setting considered in this work. To provide a more comprehensive comparison with the existing approach, as suggested, we computed additional metrics (**PA-MPJPE** and **PCKs**) as shown in Tables B and C for our method and will include them in the revision.
>
> ### Table B: Single-person pose estimation on Person-in-WiFi 3D dataset
> | Method | Params (M) | MPJPE | PA-MPJPE ★ | MPJDLE (h) | MPJDLE (v) | MPJDLE (d) | PCK@30 ★ | PCK@20 ★ | PCK@10 ★ |
> |--------|-----------|--------|-----------|-------------|-------------|-------------|---------|---------|---------|
> | MetaFi++ | 26.5 | 116.40 | 72.40 | 56.94 | 61.01 | 46.36 | 75.31 | 60.45 | 32.91 |
> | DT-Pose | 4.8 | 92.70 | 60.09 | 41.66 | _50.55_ | _38.60_ | 81.95 | 71.78 | 50.35 |
> | HPE-Li | **0.8** | 117.70 | 69.90 | 53.64 | 67.19 | 50.06 | 75.27 | 60.35 | 33.78 |
> | Person-in-WiFi 3D | 48.2 | _91.77_ | _64.51_ | _37.84_ | 51.97 | 45.00 | **94.96** | **86.20** | _59.70_ |
> | **WiFi-Mamba (Ours)** | _2.14_ | **76.75** | **62.85** | **32.11** | **47.77** | **31.41** | _86.29_ | _78.07_ | **61.09** |
>
> ### Table C: Performance comparison between baseline
> | Metric | Person-in-WiFi 3D |  |  | WiFi-Mamba (Ours) |  |  |
> |--------|-------------------|--|--|--------------------|--|--|
> |        | 1P                | 2P | 3P | 1P               | 2P | 3P |
> | MPJPE       | 91.77 | 121.04 | 144.24 | **76.75** | **89.08** | **104.81** |
> | **PA-MPJPE ★**  | 64.51 | 68.91  | 61.90  | **62.85** | **67.55** | **66.31**  |
> | MPJDLE (h)  | 37.84 | 65.75  | 61.41  | **32.11** | **48.76** | **36.93**  |
> | MPJDLE (v)  | 51.97 | 48.43  | 58.05  | **47.77** | **37.92** | **47.58**  |
> | MPJDLE (d)  | 45.00 | 55.58  | 91.21  | **31.41** | **38.43** | **62.54**  |
> | **PCK@30 ★**    | 94.96 | 77.18  | 65.66  | **86.29** | **79.44** | **73.44**  |
> | **PCK@20 ★**    | 86.20 | 60.30  | 40.00  | **78.07** | **68.64** | **58.25**  |
> | **PCK@10 ★**    | 59.70 | 25.04  | 14.51  | **61.09** | **47.67** | **32.82**  |
>
> ★ Newly added metrics as suggested by the reviewer.

---

### Decision · Program_Chairs · 2026-04-30

**Decision:**

Accept (regular)

**Comment:**

This paper proposes an SSM-based architecture for WiFi-based pose estimation. The paper received three Weak Accept and one Accept rating. The reviewers initially raised some concerns, specifically related to the paper's soundness and clarity of presentation, however all the major issues were quickly resolved after the replies from the authors. Given the unanimous accept ratings from four knowledgable reviewers, the AC finds no reason to overturn the reviewers' decision. With that being said, the AC would encourage the authors to consider all comments of the reviewers when preparing the final version of the paper, and include all the updates, clarifications and additional experiments they promised in their rebuttal.